# Anti-Viral and Immunomodulatory Properties of Propolis: Chemical Diversity, Pharmacological Properties, Preclinical and Clinical Applications, and In Silico Potential against SARS-CoV-2

**DOI:** 10.3390/foods10081776

**Published:** 2021-07-31

**Authors:** Nermeen Yosri, Aida A. Abd El-Wahed, Reem Ghonaim, Omar M. Khattab, Aya Sabry, Mahmoud A. A. Ibrahim, Mahmoud F. Moustafa, Zhiming Guo, Xiaobo Zou, Ahmed F. M. Algethami, Saad H. D. Masry, Mohamed F. AlAjmi, Hanan S. Afifi, Shaden A. M. Khalifa, Hesham R. El-Seedi

**Affiliations:** 1School of Food and Biological Engineering, Jiangsu University, Zhenjiang 212013, China; nermeen.yosri@science.menofia.edu.eg (N.Y.); guozhiming@ujs.edu.cn (Z.G.); zou_xiaobo@ujs.edu.cn (X.Z.); 2Department of Chemistry, Faculty of Science, Menoufia University, Shebin El-Kom 32512, Egypt; reemghoniem2222@gmail.com (R.G.); omarkhattab500@gmail.com (O.M.K.); ayasabry816@yahoo.com (A.S.); 3Department of Bee Research, Plant Protection Research Institute, Agricultural Research Centre, Giza 12627, Egypt; aidaabd.elwahed@arc.sci.eg; 4Computational Chemistry Laboratory, Chemistry Department, Faculty of Science, Minia University, Minia 61519, Egypt; m.ibrahim@compchem.net; 5Department of Biology, College of Science, King Khalid University, Abha 9004, Saudi Arabia; hamdony@yahoo.com; 6Department of Botany & Microbiology, Faculty of Science, South Valley University, Qena 83523, Egypt; 7Alnahalaljwal Foundation Saudi Arabia, P.O. Box 617, Al Jumum 21926, Saudi Arabia; ahmed@alnahalaljwal.com.sa; 8Department of Plant Protection and Biomolecular Diagnosis, Arid Lands Cultivation Research Institute (ALCRI), City of Scientific Research and Technological Applications, New Borg El-Arab City, Alexandria 21934, Egypt; saad.masry@adafsa.gov.ae; 9Abu Dhabi Agriculture and Food Safety Authority (ADAFSA), Al Ain 52150, United Arab Emirates; 10Pharmacognosy Group, College of Pharmacy, King Saud University, Riyadh 11451, Saudi Arabia; malajmii@ksu.edu.sa; 11Food Research Section, R&D Division, Abu Dhabi Agriculture and Food Safety Authority (ADAFSA), Abu Dhabi P.O. Box 52150, United Arab Emirates; hanan.afifi@adafsa.gov.ae; 12Department of Molecular Biosciences, Stockholm University, The Wenner-Gren Institute, SE-106 91 Stockholm, Sweden; 13International Research Center for Food Nutrition and Safety, Jiangsu University, Zhenjiang 212013, China; 14Division of Pharmacognosy, Department of Pharmaceutical Biosciences, Uppsala University, Biomedical Centre, P.O. Box 591, SE 751 24 Uppsala, Sweden

**Keywords:** propolis, chemical constituents, antiviral, immunomodulatory, clinical applications, SARS-CoV-2, molecular docking

## Abstract

Propolis, a resin produced by honeybees, has long been used as a dietary supplement and folk remedy, and more recent preclinical investigations have demonstrated a large spectrum of potential therapeutic bioactivities, including antioxidant, antibacterial, anti-inflammatory, neuroprotective, immunomodulatory, anticancer, and antiviral properties. As an antiviral agent, propolis and various constituents have shown promising preclinical efficacy against adenoviruses, influenza viruses, respiratory tract viruses, herpes simplex virus type 1 (HSV-1) and type 2 (HSV-2), human immunodeficiency virus (HIV), and severe acute respiratory syndrome coronavirus 2 (SARS-CoV-2). Over 300 chemical components have been identified in propolis, including terpenes, flavonoids, and phenolic acids, with the specific constituent profile varying widely according to geographic origin and regional flora. Propolis and its constituents have demonstrated potential efficacy against SARS-CoV-2 by modulating multiple pathogenic and antiviral pathways. Molecular docking studies have demonstrated high binding affinities of propolis derivatives to multiple SARS-CoV-2 proteins, including 3C-like protease (3CL^pro^), papain-like protease (PL^pro^), RNA-dependent RNA polymerase (RdRp), the receptor-binding domain (RBD) of the spike protein (S-protein), and helicase (NSP13), as well as to the viral target angiotensin-converting enzyme 2 (ACE2). Among these compounds, retusapurpurin A has shown high affinity to 3CL^pro^ (ΔG = −9.4 kcal/mol), RdRp (−7.5), RBD (−7.2), NSP13 (−9.4), and ACE2 (−10.4) and potent inhibition of viral entry by forming hydrogen bonds with amino acid residues within viral and human target proteins. In addition, propolis-derived baccharin demonstrated even higher binding affinity towards PL^pro^ (−8.2 kcal/mol). Measures of drug-likeness parameters, including metabolism, distribution, absorption, excretion, and toxicity (ADMET) characteristics, also support the potential of propolis as an effective agent to combat COVID-19.

## 1. Introduction

Propolis is a natural wax-like resin produced by honeybees (*Apis mellifera* L.) consisting of salivary secretions, wax, pollen, and various plant materials. Honeybees use propolis as a cement (bee glue) to seal cracks or open spaces in beehives, thereby preventing invasion by parasites and helping to maintain appropriate internal temperature and humidity [1,2]. The name propolis, from the Greek *pro* for “in defense” and *polis* for “city”, reflects its importance for preventing diseases and parasites from entering the hive and inhibiting putrefaction, fungal growth, and bacterial growth [3]. The beneficial effects of propolis on human health were recognized thousands of years ago, with reports of use in folk medicine dating back to the ancient Egyptians, Greeks, and Romans [4]. In the 17th century, the London pharmacopoeias listed propolis as an official drug [5], further highlighting the ubiquity of propolis as a disease treatment throughout the centuries. In folk medicine, propolis is used for the management of airway disorders and cutaneous-mucosal infection by bacteria and viruses [6]. In some Asian, European, and South American countries, propolis is still used to make health drinks [7]. Additionally, it used in toothpaste and mouthwash preparations for treating gingivitis, cheilitis, and stomatitis [8,9].

More rigorous laboratory investigations have documented a wide range of biological activities, such as antiseptic, anti-inflammatory, antioxidant, antibacterial, antimycotic, antifungal, antiulcer, anticancer, wound-healing, and immunomodulatory properties [10,11]. Propolis has shown antiviral activity in vitro and (or) in animal models against several DNA and RNA viruses, such as herpes simplex virus type 1 (HSV-1), an acyclovir resistant HSV1 mutant, herpes simplex virus type 2 (HSV-2), adenovirus type 2, vesicular stomatitis virus (VSV), poliovirus type 2 (PV-2) [12,13], avian influenza virus (H7N7) [14], human rhinoviruses (HRVs) [15], influenza viruses A/HlNl and A/NH3N2 [16], classic coronaviruses [17], and severe acute respiratory syndrome coronavirus 2 (SARS-CoV-2) [18,19,20]. In general, propolis consists of 50% resin, 30% wax, 10% essential oils, 5% pollen, 2% mineral salts, and an array of nutrients (intermediate metabolites) and bioactive polyphenols, mainly flavonoids, phenolic acids, and various ester and cinnamic acid derivatives [21,22]. Among these include several well described antiviral and immunomodulatory compounds such as kaempferol, *p*-coumaric acid, apigenin, artepillin C, caffeic acid, and caffeic acid phenyl ester [15,23,24].

The main aim of this review is to highlight the potential of propolis and its various constituents and/or derivatives as antiviral and immunomodulatory drugs against infectious diseases, including COVID-19 caused by SARS-CoV-2. Molecular docking analyses have identified over 40 propolis-derived compounds with strong binding affinity to various SARS-CoV-2 proteins and the human viral receptor. Moreover, measurements of drug-likeness parameters such as absorption, distribution, metabolism, excretion, and toxicity (ADMET) further support some of these agents as potential anti-SARS-CoV-2 drug candidates, warranting more extensive preclinical and clinical investigation.

## 2. Ethnopharmacology

Since ancient times, propolis has been employed by many cultures as a dietary supplement and folk remedy for improving health and managing disease [23,24]. The use of propolis in folk medicine can be traced back to at least 300 BC [25]. The Egyptians are considered the first peoples to use propolis for wound treatment and as an embalming agent [26,27]. In addition, Greek physicians such as Hippocrates, Dioscorides, and Galen; the Roman natural philosopher Pliny the Elder; and Inca healers utilized propolis as an antiseptic, antipyretic disinfectant for cutaneous and buccal infections and wound treatment. Propolis-based treatments were also in wide use in Europe during the 17th century to treat colds, wounds, rheumatism, heart disease, and diabetes [21,28]. According to Hippocrates, propolis may be used to improve health or prevent disease, including gastrointestinal disorders such as gastritis and gastric ulcer [29]. Arabs and Persians also used propolis as a disease treatment and cleansing agent [30].

All over the world, propolis has been used in traditional and folk medicine to prevent and treat many ailments, i.e., colds, wounds, rheumatism, heart disease, and diabetes [28]. Other documented uses include treatment of pharyngitis as well as wounds [31]. Brazilian green propolis is used as an anti-inflammatory, antibacterial, and antiulcer treatment in traditional medicine [32]. Administration for the treatment of abscesses and canker sores as well as wounds has also been reported [33]. During the Anglo-Boer War and Second World War, some physicians used propolis to promote tissue regeneration as well as for wound healing and treatment of tuberculosis, lung inflammation, and malnutrition [30]. Propolis with Ashwagandha (*Withania sominifera*) is used in some traditional medicine systems to boost immune function and prevent or cure various ailments [34].

## 3. Chemical Composition of Propolis

Propolis is rich in polyphenolic compounds, primarily flavonoids, cinnamic acids, and esters (Figure 1, Figure 2 and Figure 3). To date, more than 300 compounds have been isolated and identified from propolis extracts such as benzoic acid and derivatives, benzaldehyde derivatives, aliphatic hydrocarbons, saccharides, vitamins, nicotinic acid, pantothenic acid, chalcones, dihydrochalcones, amino acids, esters, minerals, enzymes, ketones, waxy acids, alcohols, and fatty acids [35]. The active constituents of propolis are diverse and vary according to the local plant species [22]. For instance, six flavonoids were isolated and identified in propolis from the southern Urals (Bashkiria): (2*S*)-5-hydroxy-7-methoxyflavanone (pinostrobin), (2*S*)-5,4′-dihydroxy-7-methoxyflavanone (sakuranetin), (2*S*)-5-hydroxy-7,4′-dimethoxyflavanone (sakuranetin-4′-methyl ether), (2*S*,3*R*)-3-acetoxy-5,7-dihydroxyflavanone (pinobanksin-3-acetate), 5,7 dihydroxyflavone (chrysin), and 5-hydroxy-7,4′-dimethoxyflavone (apigenin-7,4′ dimethyl ether) (Figure 1) [36].

Furthermore, two flavonoids (pinocembrin and chrysin) (Figure 1), *trans*-cinnamic acid, and four phenolic cinnamic acid (caffeic acid, *p*-coumaric acid, ferulic acid, and *m*-coumaric acid) (Figure 2), in addition to many volatile compounds, have been identified by high-pressure liquid chromatography with UV detector (HPLC-UV), nuclear magnetic resonance (NMR), and gas chromatography–mass spectrometry (GC–MS) from two propolis samples from two apiaries with different geographical locations in Italy [1]. From 15 Brazilian green propolis samples of different geographically area and botanical source, 47 compounds were tentatively characterized using HPLC-DAD-MS/MS and NMR, including prenylated phenylpropanoids (drupanin, capillartemisin A, 2,2-dimethylchromene-6-propenoic acid, artepillin C, and baccharin) and (*E*)-2,3-dihydroconiferyl *p*-coumarate; flavonoids and isoflavonoids (pratensein, violanone, formononetin, vestitone, and biochanin A) (Figure 1); di- and triterpenoids, as well new compounds; and (*E*)-3-[4-hydroxy-3-(2-hydroxy-3-methylbut-3-en-1-yl)-5-(3-methylbut-2-en-1-yl)phenyl] propenoic acid, among other constituents (Figure 3) [2]. The constituent profile also varies according to extraction method. Saito et al. compared conventional (Soxhlet) extraction using ethanol, methanol, water, or hexane as the solvent to supercritical CO_2_ extraction for two different types of green and red propolis from Brazil and found that methanolic extraction provided superior yield for both propolis types, while fractionation of red propolis ethanolic extract using supercritical CO_2_ yielded mixtures with much higher flavonoid content than the original extract [37].

Dudoit et al. identified 12 distinct compounds (liquiritigenin, calycosin, calycosin (isomer), luteolin, isoliquiritigenin, formononetin, (3*S*)-vestitol, (3*S*)-neovestitol, retusapurpurin A, medicarpin, retusapurpurin A (isomer), and biochanin A) in a Brazilian red propolis sample using HPLC-MS and TLC-bioautography (Figure 1). (3*S*)-Vestitol, (3*S*)-neovestitol, and medicarpin were the major components, accounting for 45% of all surface-based chromatographic peaks detected [38]. Picolotto et al. identified five additional flavonoid compounds (biochanin A, daidzein, formononetin, isoliquiritigenin, and liquiritigenin) in red propolis samples collected from Alagoas State in northeastern Brazil using ultrafast liquid chromatography-electrospray ionization-microTof mass spectrometry (UFLC-ESI-QTOF) [26]. An HPLC study of an Italian propolis sample found that the main active constituents were galangin (42.25 mg/g dry propolis, retention time (RT) =12.87 min), pinocembrin (27.30 mg/g dry propolis, RT = 10.61 min), and caffeic acid phenethyl ester (CAPE) (11.21 mg/g dry propolis, RT = 11.86 min) [31].

Recently, a multi-dynamic extraction system (RP-HPLC–PDA–ESI–MS^n^) was used to identify quercetin, pinobaskin, apigenin, chrysin, pinocembrin, and galangin in nine samples of propolis gathered separately from three different regions of Europe, America, and Asia [22,39]. Various additional extraction methods such maceration, ultrasonic extraction, and microwave extraction have also shown variable efficacy for the isolation of propolis components. For instance, ultrasonic extraction achieved a higher yield of *p*-coumaric acid (271.65 mg/g propolis) than microwave extraction or maceration. Other compounds identified by ultrasonic extraction included apigenin, caffeic acid, chrysin, galangin, isorhamnetin, kaempferol, luteolin, myricetin, pinocembrin, rutin, and quercetin [40].

## 4. Anti-Viral Activity

Propolis has long been used to treat viral infections and more recently tested for efficacy against SARS-CoV-2, the causative pathogen of COVID-19 [41]. Many disease-causing viruses are unresponsive to currently available antiviral drugs and may also evolve into more drug- and vaccine-resistant strains. Thus, it is critical to identify novel candidate antivirals, particularly from natural sources; as such compounds tend to have good safety profiles.

Herpes simplex virus (HSV) types 1 and 2 are believed to be the most prevalent human viral pathogens. HSV-1 primarily infects oral epithelial tissues, leading to watery blisters on the skin or mucosa, while HSV-2 generally infects the genital mucosa and is sexually transmitted. Acyclovir is one of the main antiviral treatments, but resistant strains are common [42]. South Turkey (Hatay) propolis at 25, 50, and 100 µg/mL was reported to suppress the replication of both HSV-1 and HSV-2 with no toxicity on infected cells, and this antiviral effect was synergetic with acyclovir (positive control) [43]. An ethanolic extract of Brazilian green propolis also demonstrated significant in vitro antiviral activity against acyclovir-resistant HSV by inhibiting replication at an early stage of infection [44]. In addition, a Brazilian hydroalcoholic brown propolis extract (HPE) was reported to protect against vaginal lesions and to reduce epidermal and dermal inflammation induced by HSV-2 in female BALB/c mice (Table 1) [45].

Moreover, aqueous and ethanolic extracts prepared from propolis showed promising antiviral efficacy against HSV-1 infection of RC-37 cells as evidenced by a plaque formation assay. These therapeutic effects may arise from masking of viral proteins necessary for adsorption or entry into host cells [13]. Aqueous and ethanolic propolis extracts also reduced HSV-2 proliferation with efficacy comparable to acyclovir when applied at different intervals during the viral infection cycle, again likely by masking viral molecules responsible for entrance or adsorption into host cells [46].

Another study conducted on propolis originating from Canada reported that propolis had significant virucidal effects against HVS-1 and -2 and also interfered with HVS-2 entry into Madin-Darby bovine kidney (MDBK) cells [47]. Moreover, French propolis proved effective against different strains of PV-2, VSV, and adenovirus type 2 (Adeno-2), as well as against HSV-1, HSV1-R, and HSV-2, with a 30 µg/mL dose showing the greatest efficacy for inhibiting viral replication [12].

Thus, propolis may inhibit HSV infection through multiple mechanisms, including virucide, inhibition of replication, and blockade of host entry.

Propolis has also demonstrated in vitro antiviral efficacy against human immunodeficiency virus (HIV), the causative pathogen for acquired immune deficiency syndrome (AIDS). According to the World Health Organization (WHO), Africa is the most severely affected region, accounting for nearly two thirds of all current HIV cases worldwide. In fact, nearly 1 in every 25 African adults (4.1%) is currently living with HIV. While there are several antiretroviral drugs available for HIV control, these have serious side effects such as lipodystrophy [54], necessitating the development of safer alternatives. Propolis extracts from different geographic regions of Brazil and China reduced HIV-1 infectivity by 85% in cultures of CD4+ cells and by 98% in cultures of microglial cells both by inhibiting viral entry and by suppressing reverse transcriptase activity [51]. Additionally, eight compounds isolated from Brazilian propolis, including four triterpenoids (melliferone, moronic acid, anwuweizonic acid, and betulonic acid) and four aromatic compounds (4-hydroxy-3-methoxypropiophenone, 4-hydroxy-3-methoxybenzaldehyde, 3-(3,4-dimethoxyphenyl)-2-propenal, and 12-acetoxytremetone), showed anti-HIV-1 activity in H9 lymphocytes (Table 2) [55].

The ethanolic extract of GH 2002 propolis demonstrated significant antiviral efficacy against varicella zoster virus (VZV), with an IC_50_ of 64 µg/mL as assessed by a plaque reduction assay. This antiviral effect was detected when propolis extract was added at different times during the viral infection cycle. Furthermore, the extract enhanced the inhibitory effect of acyclovir on viral DNA polymerase during VZV replication. These findings again suggest that bioactive components in propolis can both mask viral proteins, thus interfering with entry into host cells, and suppress viral replication [49].

As mentioned, antiviral efficacy depends on the extraction method. A study comparing the antiviral properties of green and red Brazilian propolis extract prepared using two different ultrasonic methods and maceration found that both ultrasonic extracts demonstrated greater activity against bacteriophages MS2 and Av-08 than the maceration extract, while the maceration extract of red propolis was more active than the corresponding green extract for damaging the viral cell membrane and inhibiting polymerase activity [50]. Recently, a Euro-Asian poplar propolis extract treatment for 4 days was found to inhibit H1N1 influenza virus infection of MDCK cells by suppressing both viral growth and neuraminidase (NA) activity (Table 1) [31]. Two bioactive compounds isolated from Brazilian propolis, kaempferol and *p*-coumaric acid, were tested for inhibition of HeLa cell infection by three human rhinoviruses, HRV-2, HRV-3, and HRV-4. Kaempferol is the most bioactive compound, with IC_50_ values of 7.3, 11.9, and 12.9 µM toward HRV-2, HRV-3, and HRV-4, respectively, while *p*-coumaric acid shows the lowest antiviral activity at IC_50_ values of 371.2, 454.5, and 604.3 µM in comparison with ribavirin (positive control). Findings indicate that kaempferol and *p*-coumaric acid may block or reduce the entry of the viruses into the host cells, in order to preserve the cells from virus replication [15].

Another study on the antiviral effect of Mexican propolis against MDBK cell monolayer infection by pseudo rabies virus (PRV) found that treated cells exhibited an electron-dense layer on the cell membrane that prevented viral entry [53]. Another study of 13 ethanolic extracts from South Brazilian propolis identified four with significant anti-influenza virus activity in vitro and subsequently found antiviral efficacy in vivo after oral administration to infected mice (3 times daily/7 days), with 10 mg/kg showing the greatest therapeutic effect [52].

Other work aimed to compare the effect of three samples, propolis, *Baccharis dracunculifolia* (extract and essential oil), and some isolated compounds (caffeic and cinnamic acids), on poliovirus type 1 (PV1). Three protocols (pre-, simultaneous, and post-treatments) were used for evaluating the effects on the virus. For propolis, a high inhibition percentage both in simultaneous and post-treatment was recorded. Propolis partially affects both in viral cell entry and cell replication steps in the viral cycle or leads to RNA degradation before the entry of virus to cells [48].

In summary, various bioactive compounds have been identified in Brazilian propolis extract, including antiviral agents effective against different strains of the influenza virus. Three compounds, apigenin, kaempferol, and coumaric acid, were shown to significantly inhibit the infection of MDCK cells by suppressing the post adsorption and invasion stages of viral replication [56]. In general, the antiviral activities of propolis are mediated by flavonoids and other phenolic acids. These active constituents have different modes of action, such as the formation of complexes with viral proteins required for infection (masking), formation of an electron-dense layer on the cell membrane, directly damaging viral envelope proteins, and promoting viral destruction within the cell (Figure 4).

## 5. Propolis as a Treatment for COVID-19

COVID-19 is a pandemic disease caused by the recently discovered SARS-CoV-2, the seventh known member of the coronavirus family infectious to humans (after SARS coronavirus and Middle East respiratory syndrome (MERS) coronavirus) [58]. The epidemiological burden of COVID-19 is currently a major healthcare challenge throughout the world, as SARS-CoV-2 is readily transmitted from human to human via airborne microdroplets generated during coughing, talking, or sneezing. In addition, SARS-CoV-2 can be transmitted by touching a contaminated surface and then touching the nose, mouth, or eyes [59,60]. While many drugs have been screened for efficacy against SARS-CoV-2 infection, no antiviral agent has yet proven broadly efficacious [61]. However, several natural product derivatives have shown promise as effective non-toxic antiviral agents [62]. Potential therapeutic agents may include honeybee products in addition to propolis, such as honey, royal jelly, bee venom, wax, bee pollen, and bee bread, as all have demonstrated antimicrobial, antifungal, anti-inflammatory, and (or) antiviral properties under certain conditions [63]. Propolis has also shown promising broad spectrum antiviral effects in vitro and in vivo against influenza virus, human respiratory syncytial and coronaviruses, rotavirus, and human rhinovirus, among others, suggesting potential efficacy against coronaviruses [16,64,65].

The potential efficacy of five propolis-derived flavonoids was recently evaluated in vitro on different DNA and RNA viruses, including coronaviruses, using the viral plaque reduction technique. Acacetin and galangin had no effect on either the infectivity or replication of any of the viruses tested, but chrysin and kaempferol were highly effective in inhibiting replication, and quercetin was active against infectivity and replication at higher concentrations [17]. Refaat et al. investigated the in vitro effects of crude Egyptian propolis extract and a propolis liposome preparation on SARS-CoV-2 3CL-like protease, S1 spike protein, and viral replication by RT-PCR. Liposomes inhibited SARS-CoV-2 3CL protease activity with an IC_50_ of 1.183 ± 0.06 µg/mL, while the crude propolis extract inhibited 3CL protease activity with an IC_50_ of 2.452 ± 0.11 µg/mL, values comparable to Remdesivir [18]. Sulawesi propolis and its components glyasperin A, broussoflavonol F, and sulabiroins A also inhibited SARS-CoV-2 3C-like protease activity and interacted with the protease catalytic sites His^41^ and Cys^145^, with docking scores of −7.8, −7.8, and −7.6 kcal/mol, respectively [19]. Similarly, Hashem et al. evaluated the in silico inhibitory activity of six selected compounds present in propolis, 3-phenyllactic acid, CAPE, lumichrome, galangin, chrysin, and caffeic acid, against SARS-CoV-2 3CL^pro^ and found that all six showed good docking scores, with the most potent being CAPE (−6.383 kcal/mol), chrysin (−6.097), and galangin (−6.295) [20].

Other studies have found that propolis is able to inhibit the activity of P21 (RAC1) Activated Kinase 1 (PAK1), a major “pathogenic” kinase in several diseases/disorders, including inflammation, cancer, malaria, and pandemic viral infections such as HIV, influenza, and COVID-19 [66]. Additionally, CAPE was found to bind and inhibit SARS-CoV-2 transmembrane protease serine 2 to a degree comparable with Camostat mesylate as evidenced by molecular docking and molecular dynamics (MD) simulations [34]. Moreover, quercetin alone and in conjunction with vitamin C was predicted to suppress SARS-CoV-2 infection by binding to 3C-like protease (3CL^pro^) [67,68]. A pilot randomized clinical study assessed the potential efficacy of Brazilian green propolis (400 or 800 mg/day orally or via nasoenteral tube) against SARS-CoV-2 (NCT04480593). In addition, it supported the idea that propolis may be an effective agent to combat coronavirus-induced fibrosis in the lungs [65].

## 6. Immunomodulatory Activity

Although propolis has been mentioned as an immunomodulatory agent for centuries, little was known about its action until the 1990s. In the last decade, however, new and interesting articles have been published, contributing greatly to this field of research [69]. The immunomodulatory activity of propolis standard extract in allergic asthma was investigated by Piñeros [70] (Figure 5). The chronic inflammatory disease is mediated by Th2 inflammation and an increased number of CD4^+^ T cells, which produces an excess of cytokines including interleukin-4 (IL-4), IL-5, and IL-13. It is also characterized by eosinophilic infiltration and mast cell activation [71,72]. Ovalbumin (OVA)-induced allergy model animals were treated daily by gavage with 150 mg/Kg of propolis for 17 days. Propolis treatment reduced pulmonary Th2 inflammation and decreased eosinophils infiltration as well as IL-5 levels in Bronchoalveolar lavage fluid (BALF). Propolis also induced the differentiation and frequency of myeloid-derived suppressor cells (MDSC) and CD4^+^ Foxp3^+^ regulatory T cells [70]. These findings are consistent with a study by Sy et al., reporting that low and high doses of propolis aqueous extract (65 mg/kg and 325 mg/kg body weight) decreased BALF IL-5 concentration, IL-6 and IL-10 production by splenocytes, and the serum levels of immunoglobulin E (IgE) and immunoglobulin G (IgG) antibodies [73].

The immunosuppressive properties of propolis have also been investigated in models of rheumatoid arthritis (RA). Dietary administration of propolis ethanolic extract (6.7 and 20 mg/g) was found to reduce the severity of this autoimmune disease in vivo by inhibiting production of IL-17 [75], a pro-inflammatory cytokine produced by Th cells (Th17 cells) strongly implicated in RA pathogenesis (e.g., joint inflammation and destruction of bone and cartilage) [76]. As such, targeting Th17 cells and targeting the IL-17 signaling pathway are potentially effective strategies for RA treatment, and indeed, such treatments are currently under investigation [77]. Okamoto et al. reported that Brazilian propolis suppressed Th17 cell activity in vitro at 12.48 µg/mL by inhibiting the IL-6-induced phosphorylation of signal transducer and activator of transcription 3 (STAT3), a key transcription factor driving Th17 cell differentiation. In addition, Th17 cell differentiation induced by transforming growth factor-*β* (TGF-*β*) plus IL-16 was downregulated by propolis in RA model animals, while propolis induced no detectable cellular toxicity at concentrations up to 96 µg/mL [78]. The propolis-derived compound CAPE has also been reported to suppress autoimmune uveoretinitis. CAPE was found to hinder T cell-dependent production of chemokines and cytokines as well as of antibodies induced by interphotoreceptor retinoid binding protein (IRBP). Treatment with 200 µL CAPE also reduced serum concentrations of TNF-*α*, IL-6, interferon-*γ* (IFN-*γ*), and TNF-*α* in the retina and inhibited the transcriptional activity of nuclear factor-kappa B (NF-kB) and phospho-IkB*α*. Hence, it was concluded that the immunosuppressive activity of CAPE in uveitis is mediated by suppression of the pro-inflammatory NF-kB–cytokine pathway [79].

Propolis extracts and derivatives may also augment microbe-induced immune responses by modulating Toll-like receptor (TLR) signaling. Toll-like receptors recognize pathogen-associated molecular patterns (PAMPs), conserved molecules expressed by many microorganisms [80]. Toll-like receptor 2 (TLR2), for instance, recognizes lipoteichoic acid on Gram-positive bacteria and fungi, while TLR-4 recognizes lipopolysaccharide on Gram-negative bacteria [81]. Toll-like receptors are mainly expressed by antigen-presenting cells (APCs), including monocytes, macrophages, B cells, and dendritic cells (DCs) [82]. These cells also express human leukocyte antigen-DR isotype (HLA-DR) and cluster of differentiation 80 (CD80) molecules that present peptides to T cells, resulting in T cell activation. Additionally, TLR signal transduction may activate transcription factors controlling the expression of genes encoding chemokines, cytokines, and antimicrobial peptides [83].

Treatment of BALB/c mice with 200 mg/kg of 30% propolis ethanolic extract for three consecutive days increased expression of TLR-2 and TLR-4 by peritoneal macrophages and spleen cells and elevated the production of IL-1β and IL-6 [84]. In another study, propolis treatment of mice prevented the inhibition of TLR-2 and TLR-4 induced by 14 days of restraint stress. Additionally, real-time polymerase chain reaction (RT-PCR) revealed a significant increase in TLR gene expression in mice receiving propolis treatment without stress [81]. Propolis treatment (10, 20, and 40 µg/mL) also increased the expression of TLR-4 and CD8 by human DCs through a mechanism involving has-miR-155, resulting in enhanced bactericidal activity against *Streptococcus mutans*, and promoted the production of NF-kB, TNF-*α*, IL-6, and IL-10 [85]. Conversely, cinnamic acid (5–100 μg/mL) downregulated the expression levels of TLR-2, HLA-DR, and CD80 by human monocytes, although this treatment upregulated TLR-4. High concentrations of cinnamic acid also inhibited expression of TNF-*α* and IL-10. As TNF-*α* is known to activate monocytes and macrophages, while IL-10 inhibits these cells, Conti et al. concluded that the increase in fungicidal activity could be due to mechanisms involving other cytokines [83]. Following a similar protocol, Búfalo et al. found that caffeic acid inhibited the expression of TLR-2 and HLA-DR, while CD80 and TLR-4 were not affected. The fungicidal activity of monocytes increased, however, despite the decrease in TNF-*α* and IL-10 [86].

The in vivo antileishmanial effect of Brazilian propolis was reported for the first time by Pontin et al., that is, an administration of the hydroalcoholic extract at a dose of 1.5 mg/kg/day reduced the lesion diameter in *leishmania braziliensis* infected albino mice by 90% after 90 days of treatment. Pontin et al. pointed out that the reduction could be a result of the activation of macrophages and their phagocytic capacity [87]. Consistent with this explanation, da Silva et al. found that 5 and 10 µg/mL propolis activated macrophage phagocytic activity and in turn increased parasite interiorization. This upregulation of macrophage activity was attributed to increased TNF-*α* and reduced IL-12 signaling. Morphological changes in promastigote forms of *Leishmania* were also observed by scanning electron microscopy upon treatment [88]. Additionally, propolis was found to regulate the expression of CCL5 and IFN-γ, factors involved in the development of Th1 cells in leishmaniasis patients. Leishmaniasis is usually associated with the development of a strong Th1 response that impairs the wound healing process [89]. In another study by dos Santos Thomazelli et al., the hydroalcoholic extract showed an immunomodulatory effect on both healthy donors and American tegumentar leishmaniasis patients’ human-derived peripheral blood mononuclear cells (PBMC) in leishmaniasis models. This impact was explained by the increase in IL-4 and IL-17 and a decrease in IL-10 in a dose-dependent manner. On the other hand, nitric oxide (NO) levels remained constant [90].

Numerous studies have also suggested that propolis extracts can suppress tumor growth or promote immune-mediated tumor destruction. Benkovic et al. examined the possible synergistic effect of a water-soluble derivative of propolis (WSDP) and ethanolic extract of propolis (EEP) with the anticancer drug irinotecan in Swiss albino mice inoculated with Ehrlich ascites tumor (EAT) cells. Intraperitoneal injection of WSDP and EEP at 100 mg/kg for three days prior to 50 mg/kg irinotecan injection enhanced the antitumor efficacy and reduced the non-target cytotoxicity of irinotecan compared to irinotecan alone or the combination of irinotecan with the phenolic compounds quercetin and naringin [91]. Further investigation revealed that the decrease in irinotecan-induced non-target cytotoxicity was due to the immunomodulatory properties of propolis. Pretreatment with WSDP+EEP activated macrophages and increased the number of neutrophils in the peritoneal cavity [92]. Oršolić et al. [93] also reported that WSDP at 50 or 150 mg/kg suppressed metastasis and tumor development in mice transplanted with mammary carcinoma cells. This antimetastatic effect was associated with macrophage activation and ensuing nonspecific tumor resistance. Additionally, high levels of lymphocyte activating factor (LAF) produced by these activated macrophages increased tumor cell killing efficiency. Furthermore, WSDP significantly increased the expression of CD4^+^ and CD8^+^ by splenocytes [93]. It was concluded that the antitumor activity of WSDP is likely due to the synergistic effects of constituent polyphenolic compounds such as caffeic acid, quercetin, chrysin, and naringenin, and it was further proposed that these compounds interfere with tumor growth by enhancing apoptosis, macrophage activation, and production of pro-inflammatory cytokines such as IL-1, IL-6, IL-8, TNF-α, and NO, some of which can directly damage tumor cells, whereas others act indirectly by enhancing the activities of natural killer (NK) cells and cytotoxic T lymphocytes. Furthermore, these factors stimulate the production of complement factor C3 production and C-reactive protein, which participate in the opsonization of tumor cells [94,95,96].

Propolis was also shown to reduce the severity of Aujeszky disease when used as a vaccine adjuvant. Mice treated with 5 mg propolis extract, aluminum hydroxide Al(OH)_3_, and inactivated Suid herpesvirus type 1 (SuHV-1) demonstrated significantly higher neutralizing antibody titers than mice receiving vaccine without propolis, indicating that the adjuvant properties of propolis are associated with enhanced humoral and cellular immunity related to increased IFN-ɣ mRNA production. Moreover, expression of mRNA IFN-ɣ was even higher when propolis was conjugated with antigen [97]. Although numerous preclinical studies have shown the potential efficacy of propolis against immunological diseases, standardized quality controls and well-designed clinical trials are needed before propolis or its components can be adopted as therapeutics (Table 3) [98].

## 7. Clinical Applications of Propolis as an Antiviral and Immunomodulatory Agents

According to a recent review, six separate trials have found that propolis possesses better antiviral efficacy against herpes viruses than acyclovir (summarized in Table 4) [121]. A randomized, single-blind study involving 90 men and women diagnosed with HSV type 2 reported that a significantly greater number of patients treated with propolis ointment containing natural flavonoids (24 of 30) achieved symptom amelioration compared to patients receiving acyclovir (14 of 30) or vehicle (12 of 30) as determined by gynecologists, dermatovenerologists, or urologists, with no difference in medication-related adverse effects (Table 4) [122].

Similarly, a propolis lotion produced significantly greater healing rates than a propolis-free lotion (placebo control) following Herpes zoster virus infection (*p* < 0.001 for pain reduction at all visits, reduced new vesicles on day 7 of treatment, and greater global efficacy on the last (28th) day of treatment) with excellent skin tolerability and no allergic reactions, skin irritations, or other adverse events [123]. Holcová et al. utilized three different concentrations (0.1%, 0.5%, and 1%) of propolis special extract GH 2002 in a lip balm through a double-blind, randomized dermatological study involving 150 patients infected with Herpes labialis, and all three concentrations of propolis proved to be effective against Herpes labialis (*p* < 0.0005) for painless incrustation and local pains, but good tolerability was observed with the 0.5% concentration [125]. Another study conducted by Arenberger et al. showed the ability of propolis special extract (GH 2002) at 0.5% to treat episodes of herpes labialis virus versus 5% Acyclovir cream (*p* < 0.0001), and no allergic reactions, local irritations, or other adverse effects were observed (Table 4) [126].

Furthermore, propolis extract was evaluated in another double-blind, randomized, placebo-controlled trial, which reported the ability of propolis (200 mg three times daily for 7 days) to treat patients with dengue hemorrhagic fever virus, a faster recovery in platelet counts (*p* = 0.006), a greater decline in circulating TNF level (*p* = 0.018), and a shorter hospitalization period compared with placebo-treated patients (*p* = 0.012) (Table 4) [3].

Although propolis has plenty of biological and pharmacological properties, there is a lack of clinical reports on the effectiveness of propolis. Furthermore, some research has indicated that propolis is unsafe because it induces hypersensitivity and might induce adverse reactions such as allergic cheilitis and oral ulceration (Table 4) [128].

## 8. In Silico Drug Discovery

In seeking potential natural products to combat COVID-19 disease, the molecular docking technique was applied to portend the binding modes and affinities of 40 propolis derivatives towards five viral targets and one human target, namely 3CL^pro^, RNA-dependent RNA polymerase (RdRp), papain-like protease (PLpro), receptor-binding domain (RBD) of the spike protein (S-protein), helicase (NSP13), and human angiotensin-converting enzyme 2 (ACE2). The technical details of the employed molecular docking calculations are described in References [129,130,131,132,133,134,135,136]. In brief, the crystal structures of SARS-CoV-2 3CL^pro^ (PDB code: 6LU7) [137], RdRp (PDB code: 6M71) [138], PL^pro^ (PDB code: 6W9C) [139], RBD (PDB code: 6M0J) [140], and NSP13 (PDB code: 5RMM) [141] were opted for as templates for all molecular docking calculations. For human ACE2, the 3D structure was taken from PDB code 6M0J [140]. For target preparation, all crystallographic water molecules, ions, heteroatoms, and ligands, if existing, were removed. All missing amino acid residues were constructed with the help of Modeller software [142]. Furthermore, the protonation states of the viral and human targets were inspected utilizing the H^++^ web server, and all missing hydrogen atoms were inserted [143]. The pdbqt files for the viral and human targets were then prepared in accordance with the AutoDock protocol [144]. The chemical structures of the investigated propolis derivatives were retrieved in SDF format from the PubChem database (https://pubchem.ncbi.nlm.nih.gov, accessed on 23 March 2021). Omega2 (software version 2.5.1.4, OpenEye Scientific Software, Inc., Santa Fe, NM, USA) was applied to generate the 3D structures of the investigated compounds [145,146]. All compounds were subsequently energetically minimized using the Merck Molecular Force Field 94 (MMFF94S) implemented inside SZYBKI software [147,148]. All molecular docking calculations were conducted using AutoDock4.2.6 software [149]. The maximum number of energy evaluations (eval) and the genetic-algorithm number (GA) were set to 25,000,000 and 250, respectively. All other docking parameters were kept at their default values. The docking grid box dimensions were set to 60 Å × 60 Å × 60 Å with a spacing value of 0.375 Å to encompass the active sites of the viral and human targets. The grid center was located at the center of the binding pockets of the targets. The Gasteiger method was applied to assign the atomic partial charges of the investigated propolis derivatives [150]. To predict the pharmacokinetic properties of the identified potential anti-viral propolis derivatives, the admetSAR server (http://lmmd.ecust.edu.cn/admetsar2/, accessed on 12 April 2021) was used. The pharmacokinetic properties included Lipinski’s rule of five, absorption, distribution, metabolism, excretion, and toxicity. The results gained were analyzed and compared to the reference values of the admetSAR pharmacokinetics expectation properties [151].

The docking scores and binding features of the 40 propolis derivatives against the viral and human targets were predicted and summarized in Table 5. For comparison purposes, the corresponding data for darunavir and favipiravir were predicted. Three- and two-dimensional representations of binding modes of the most potent propolis derivatives inside the active site of the viral and human targets are depicted in Figure 6. What is interesting about the data in Table 5 is that most propolis derivatives demonstrated good binding affinities against SARS-CoV-2 and ACE2 targets. The estimated docking scores ranged from −5.5 to −9.4 kcal/mol, from −4.9 to −8.2 kcal/mol, from −4.5 to −7.5 kcal/mol, from −5.0 to −9.4 kcal/mol, and from −5.5 to −10.4 kcal/mol with 3CL^pro^, PL^pro^, RdRp, NSP13, and ACE2, respectively. For the viral RDB target, propolis derivatives manifested moderate binding affinities with docking scores ranged from −4.0 to −7.2 kcal/mol. The observed potentiality of propolis derivatives towards the SARS-CoV-2 and ACE2 targets could be attributed to their capability of forming several hydrogen bonds, hydrophobic interactions, van der Waals, and pi-based interactions with the proximal amino acid residues inside the active site of these scrutinized targets. Interestingly, retusapurpurin A demonstrated the highest binding affinities towards 3CL^pro^, RdRp, RBD, NSP13, and ACE2 with docking scores of −9.4, −7.5, −7.2, −9.4, and −10.4 kcal/mol. More precisely, retusapurpurin A forms four hydrogen bonds with the key amino acids inside the active site of 3CL^pro^, RdRp, and NSP13 (Figure 6). Three hydrogen bonds were observed between retusapurpurin A and the proximal amino acid residues of ACE2, namely ASP206 (2.23 Å) and ASN210 (2.16 and 2.24 Å) (Figure 6). However, retusapurpurin A forms only two hydrogen bonds with TYR365 (1.71 Å) and ALA366 (2.14 Å) inside the active site of RBD of S-protein (Figure 6). For the PL^pro^ target, baccharin displayed the highest binding affinities with a docking score of −8.2 kcal/mol. Eventually, baccharin exhibits three hydrogen bonds with ARG157 (2.17 Å), GLU196 (2.70 Å), and MET201 (2.20 Å) inside the active site of PL^pro^ (Figure 6). Compared to retusapurpurin A and baccharin, darunavir revealed a good binding affinity towards 3CL^pro^, ACE2, and NSP13 with docking scores of −8.2, −8.5, and −7.1 kcal/mol, respectively. On the other hand, darunavir revealed a low binding affinity of −4.4, −3.8, and −3.3 kcal/mol against RdRp, PL^pro^, and RBD of S-protein, respectively. In contrast, favipiravir manifested poor binding affinities with docking scores of −4.3, −4.0, −4.8, −4.4, −4.2, and −4.3 kcal/mol against 3CL^pro^, PLpro, RdRp, ACE2, RBD, and NSP13, respectively. The current data shed new light on the significance of retusapurpurin A and baccharin as promising SARS-CoV-2 and ACE2 inhibitors.

The pharmacokinetics, physicochemical, and toxicological properties, as summarized in Table 6, provide a quantitative analysis of what the human body performs to an administrated molecule. Based on Lipinski’s rule of five (RO5), retusapurpurin A and baccharin obey the criteria for orally active drugs.

It can be seen from the data in Table 6 that retusapurpurin A and baccharin showed a great Caco-2 permeability. Additionally, retusapurpurin A and baccharin demonstrated a great human intestinal absorption (HIA) with an estimated HIA value of ≈97% (Table 6). For the blood–brain barrier (BBB permeability), retusapurpurin A and baccharin were predicted to pass the BBB easily. Ultimately, retusapurpurin A and baccharin were non-carcinogenic and non-toxic. Overall, the predicted results displayed that the ADMET properties of retusapurpurin A and baccharin are extremely satisfying, as presented in Table 6. Consequently, retusapurpurin A and baccharin should be considered as prospective drug candidates against COVID-19.

## 9. Conclusions

Propolis is among the few natural remedies that have been utilized for centuries, and modern laboratory investigations have confirmed that the effectiveness of propolis originates from its extracts and derivatives against multiple disease models, including viral infections. These therapeutic effects are attributable to a high content with pharmacologically active molecules, mainly concentrations of bioactive flavonoids, phenolic acids, and their esters. These components have significant activities that target a myriad of pathological and reparative processes, including immune signaling pathways. The demonstrated efficacy against a wide range of human viruses has provided a rationale, and paved the way, for studies on the efficacy of propolis to be tested against SARS-CoV-2. Furthermore, 40 propolis derivatives have shown a high affinity SARS-CoV-2 proteins and the human target ACE2, with one, retusapurpurin A, demonstrating particularly potent binding to SARS-CoV-2 3CL^pro^, RdRp, ACE2, RBD, and NSP13 inhibitor. These results suggest that retusapurpurin A and other components such as baccharin are promising, and with further investigation, they could be used as potential weapons in the fight against the pandemic and as therapeutic candidates for COVID-19 treatment.

In addition to viral infection, propolis has anti-inflammatory and immunomodulatory activities. Propolis was proven to be effective through different mechanisms against several immune-mediated models of cancer and immune-related diseases, i.e., celiac disease, uveoretinitis, allergic asthma, rheumatoid arthritis, leishmaniasis, microbial infections, and cancer in vitro and in vivo. There is, however, little knowledge. While there have been few studies on the clinical efficacy of propolis and its effects on human health, further investigations are needed to determine its application. Forty propolis derivatives were investigated in in silico preparations against five SARS-CoV-2 and human ACE2 targets as anti-COVID-19 drug candidates with the help of the molecular docking technique. The binding affinities unveiled that retusapurpurin A is a potent SARS-CoV-2 3CLpro, RdRp, ACE2, RBD, and NSP13 inhibitor. However, baccharin demonstrated the highest binding affinity against SARS-CoV-2 PLpro. Moreover, drug-likeness and ADMET properties were predicted for the most potent compounds and demonstrated satisfactory pharmacokinetics, physicochemical parameters, and toxicological properties. These results suggest that retusapurpurin A and baccharin can be further investigated as convenient therapeutic treatments for COVID-19 disease, and clinical trials are feasible due to the generally good safety profiles of propolis derivatives.

## Figures and Tables

**Figure 1 foods-10-01776-f001:**
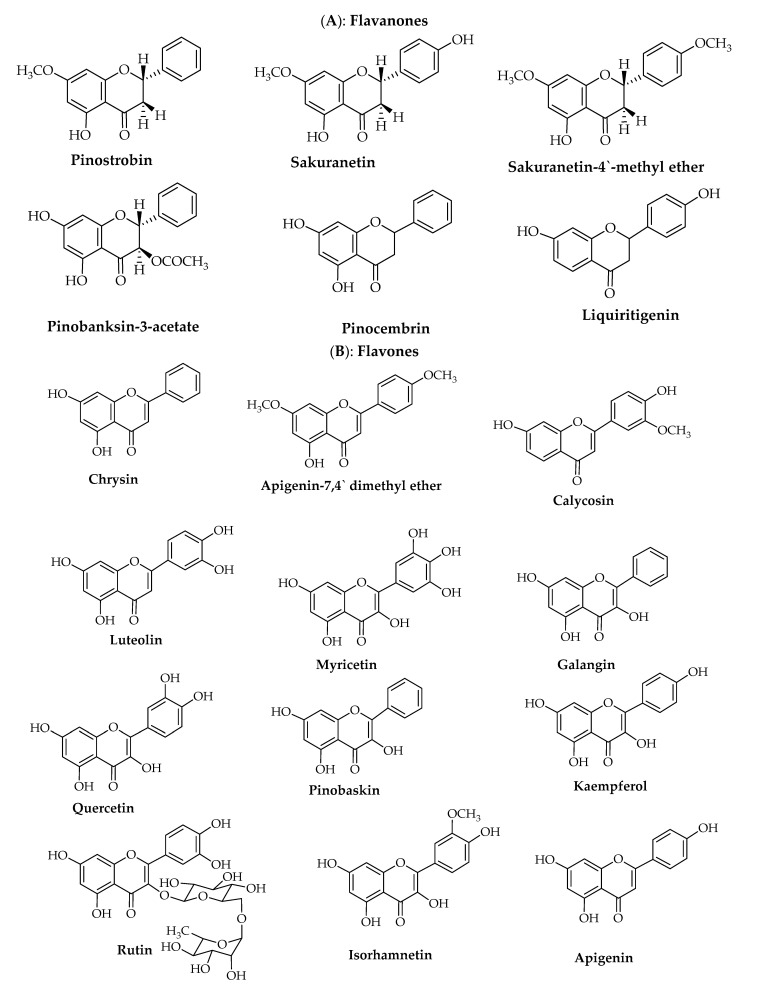
Common flavonoid compounds isolated and identified from propolis of *Apis mellifera* L.

**Figure 2 foods-10-01776-f002:**
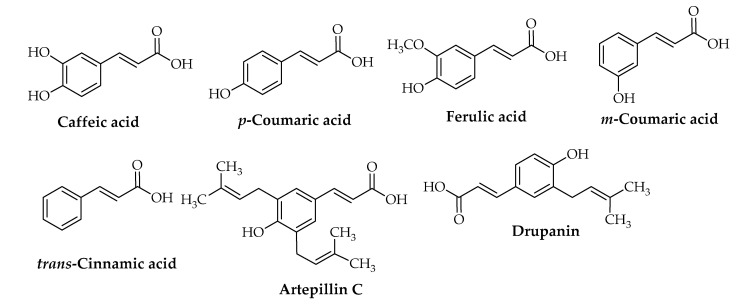
Common cinnamic acid compounds isolated and identified from propolis of *Apis mellifera* L.

**Figure 3 foods-10-01776-f003:**
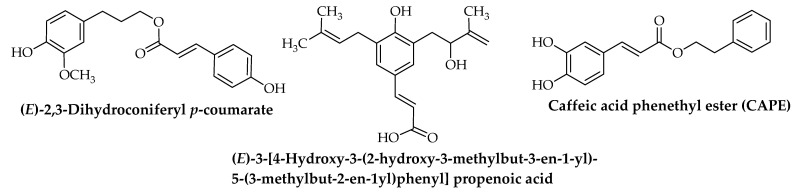
Common ester and acid compounds isolated and identified from propolis of *Apis mellifera* L.

**Figure 4 foods-10-01776-f004:**
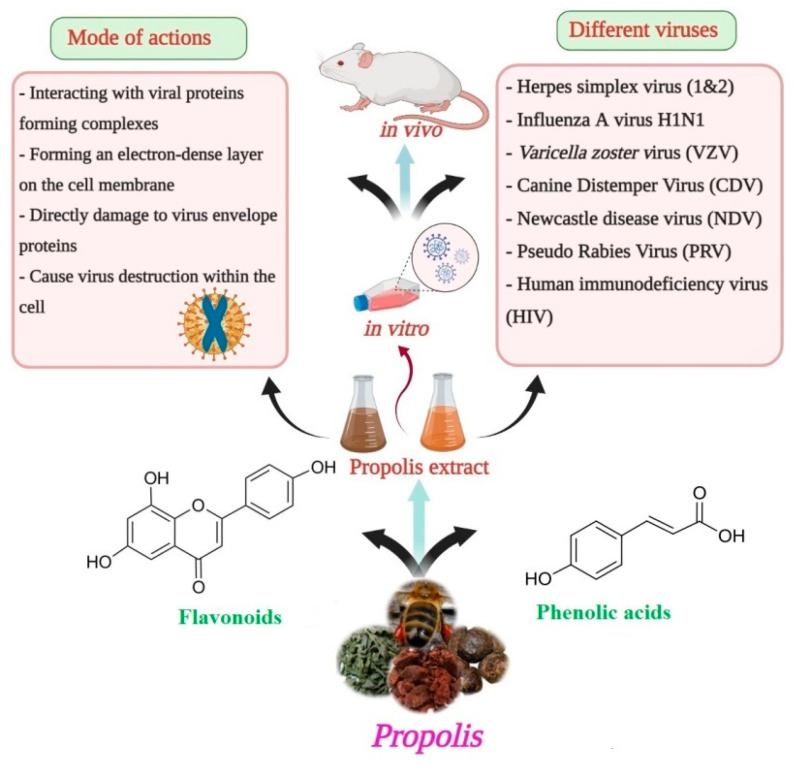
Anti-viral activity and the possible mode of action of propolis extracts/constituents against different types of viruses.

**Figure 5 foods-10-01776-f005:**
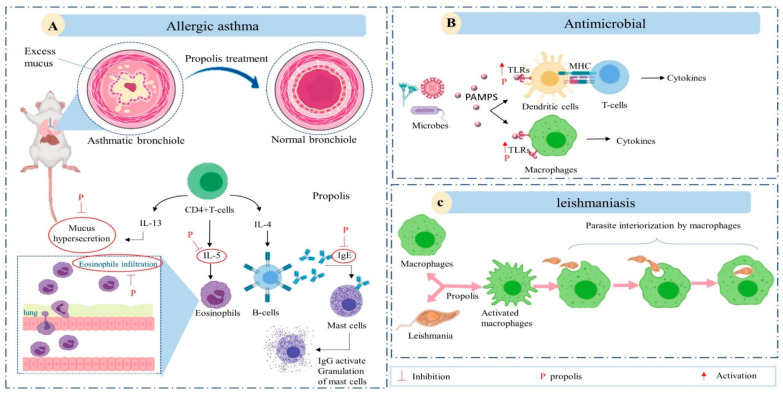
Representative scheme demonstrates some of the possible immune-mediated mechanisms of propolis in (**A**) Allergic asthma: Asthma is mediated by Th2 inflammation in which CD4 T-cells produce an excess of IL-4, IL-5, and IL13 cytokines. IL-4 triggers the production of IgG antibodies by B-cells, whereas IL-5 activates and promotes the infiltration of eosinophils to airways inducing bronchial inflammation, and IL-13 mainly participates in mucus hypersecretion [74]. Propolis inhibits IL-5 production, mucus hypersecretion, eosinophil infiltration, and IgE produced by B cells. (**B**) Antimicrobial: Propolis upregulates the production of Toll-like receptors (TLRs) which have a vital role in the recognition of PAMPs present in most microbes. TLRs signal macrophages and dendritic cells triggering the production of inflammatory cytokines. (**C**) Leishmaniasis: Propolis activates macrophages and their phagocytic capacity.

**Figure 6 foods-10-01776-f006:**
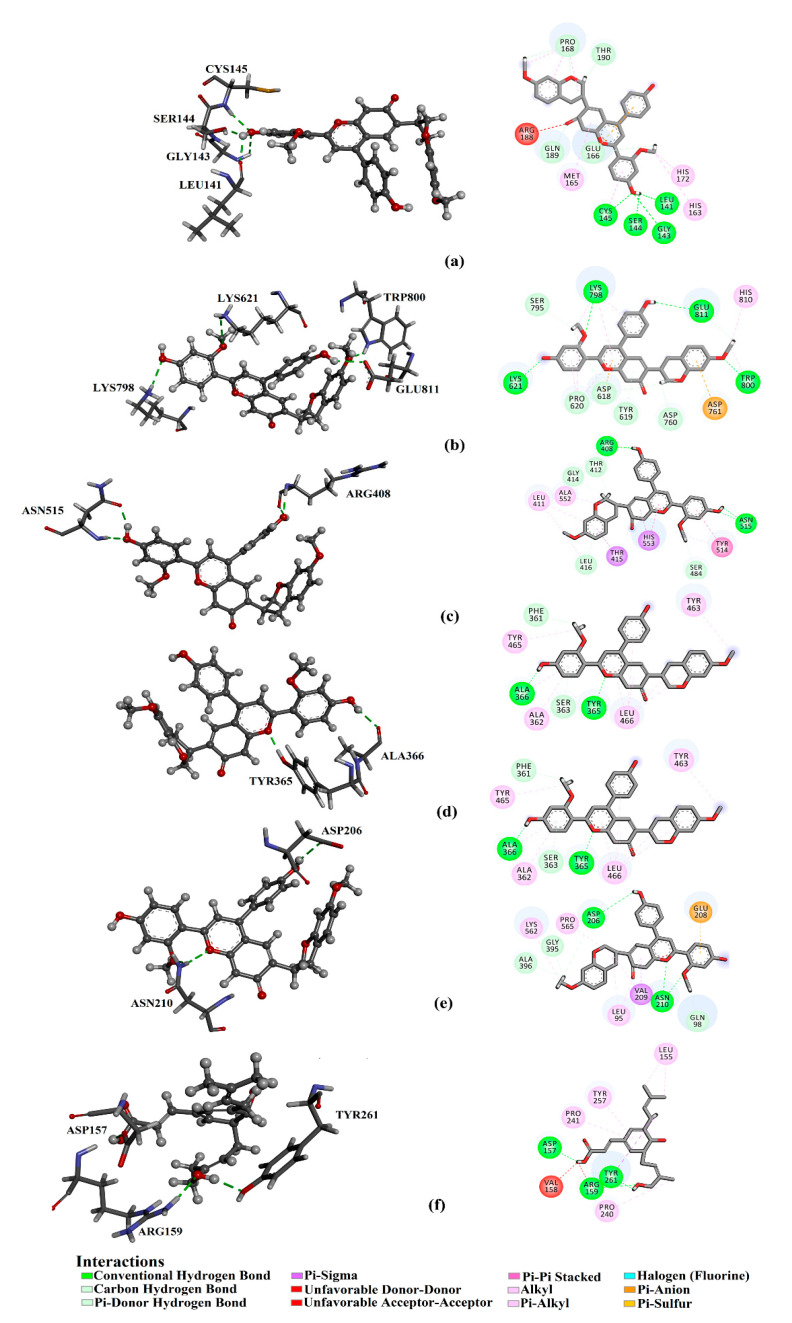
3D and 2D representations of interactions of retusapurpurin A with proximal amino acid residues of SARS-CoV-2 (**a**) 3C-like protease (3CL^pro^), (**b**) RNA-dependent RNA polymerase (RdRp), (**c**) helicase (NSP13), (**d**) receptor-binding domain (RBD) of the spike protein, and € (**e**) human angiotensin-converting enzyme 2 (ACE2), as well as (**f**) baccharin with papain-like protease (PL^pro^).

**Table 1 foods-10-01776-t001:** The effect of propolis as antiviral activity against different viruses.

Propolis Origin	Type of Extract	Antiviral Activities	References
South of Turkey (Hatay region)	70% Ethanol	Herpes simplex virus (HSV-1) Assay: MTT colorimetric and Real-Time PCR Model: HEp-2 cell culture MIC = 130 μg/mL Tested doses: 25, 50, 75, 100, 200, 400, 800, 1600 and 3200 μg/mL PC: Acyclovir MIC = 65 μg/mL NC: Cell cultures without propolis or acyclovir Mode of action: Inhibits the replication after 24 h. (in vitro)	[43]
		Herpes simplex virus (HSV-2) Assay: MTT colorimetric and Real-Time PCR Model: HEp-2 cell culture MIC = 250 μg/mL Tested doses: 25, 50, 75, 100, 200, 400, 800, 1600 and 3200 μg/mL PC: Acyclovir NC: Cell cultures without propolis or acyclovir Mode of action: Inhibits the replication after 48 h. (in vitro)	[43]
Northwestern Parana state, Brazil	70% Ethanol	Herpes simplex virus 1 (HSV-1) Assay: Attachment and penetration; attachment, virucidal, and plaque formation Model: Vero cells EC_50_ = 3.20 ± 0.14 µg/mL EC_50_ for attachment and penetration: 1.21 ± 0.14 µg/mL EC_50_ for attachment: 0.40 ± 0.07 µg/mL EC_50_ for virucidal: 3.84 ± 0.15 µg/mL PC: Acyclovir EC_50_ = 1.33 ± 0.08 µg/mL NC: Untreated cells Mode of action: Inhibits viral infection and induces virion damage. (in vitro)	[44]
Santa Flora City (RS-Brazil)	70% Ethanol	Herpes simplex virus 2 (HSV-2) Assay: Plaque reduction Model: Female BALB/c mice Dose of Pre-treatment: 50 mg/kg, once a day Dose of Post-treatment: 50 mg/kg for 5 days more PC: Not reported NC: Untreated cells Mode of action: Reduces extravaginal lesions and the histological damage caused by HSV-2 infection in vaginal tissues of animals. (in vivo and ex vivo)	[45]
Moravia, Czech Republic	- Aqueous extract (15% ethanol) - 90% Ethanol	Herpes simplex virus 1 (HSV-1) Assay: Plaque reduction Model: RC-37 cells Aqueous extract: TC_50_ (%): 0.04; IC_50_ (%): 0.0004; SI: 100 Ethanol extract: TC_50_ (%):0.0017; IC_50_ (%): 0.000035; SI: 485 PC: Heparin-Na and acyclovir NC: Untreated cells Mode of action: Mask viral compounds which are necessary for adsorption or entry into host cells. (in vitro)	[13]
Moravia, Czech Republic	- Aqueous extract (15% ethanol) - 90% Ethanol	Herpes simplex virus type 2 (HSV-2) Assay: Plaque reduction Model: RC-37 cells IC_50_% for aqueous extract: 0.0005; SI: 80 IC_50_% for ethanolic extract: 0.0004; SI: 42.5 PC: Acyclovir Inhibits replication of 98.8% NC: Untreated cells without drugs Mod of action: - Suppresses HSV multiplication; - Masks viral compounds which are necessary for adsorption or entry into host cells. (in vitro)	[46]
Canada	70% Ethanol	Herpes simplex virus 1 and 2 Assay: Virucidal assay Model: MDBK cell PPE marked effect: 3.2 mg/mL PC: Acyclovir NC: Not reported Mode of action: - Propolis had a pronounced virucidal effect against herpes simplex viruses type 1 and type 2, and also interfered with virus adsorption; - Suppresses the adsorption of HSV-1 at a broad scope of the viral inoculation. (in vitro)	[47]
Botucatu, Brazil	70% Ethanol	Poliovirus type 1 (PV1) Assay: Real-time PCR Model: HEp-2 cells - Pre-treatment: 10.9% - Simultaneous treatment: 52.2% - Post-treatment: 39.1% Propolis conc: 5, 10, 25, 50, and 100 µg PC: Cells with virus but without propolis NC: Not reported Mode of action: Causes RNA degradation before the virus entry into cells; also affects the steps of viral cycle replication into cells. (in vitro)	[48]
Rennes (France)	80% Ethanol	Poliovirus type 2 (PV), vesicular stomatitis virus (VSV), adenovirus type 2 (Adeno-2), herpes simplex Virus (HSV-1, HSV1-R, and HSV-2) Assay: Plaque reduction Model: Vero cells Dose: 30 µg/mL PC: NR NC: Without propolis Mode of action: NR (In vitro)	[12]
Moravia, Czech Republic	90% Ethanol	*Varicella zoster* virus (VZV) Assay: Plaque reduction and PCR Model: Cell line of human embryonic lung fibroblasts (LEP) IC_50_: 64 μg/mL Tested doses: 100 μg/mL PC: Acyclovir (conc: 50 μg/mL) NC: Untreated culture Mode of action: Masking viral compounds which are necessary for entry into host cells. (in vitro)	[49]
Paraná state and Alagoas state, Brazil	80% Ethanol	Enterovirus surrogates MS2 and Av-08 bacteriophage Assay: Plaque formation Tested conc.: 100, 500, and 1000 μg/mL PC: Not reported NC: Bacteriophage with 1 mL of *E. coli* O157 bacteria without the addition of the propolis extracts Mode of action: Inhibits viral polymerase and the binding of viral nucleic acid or capsid proteins. (in vitro)	[50]
Tavarnelle Val di Pesa, Firenze, Italy	80% Ethanol	Influenza A virus H1N1 Model: Madin-Darby canine kidney cells (MDCK) Anti-neuraminidase IC_50_ (µg/mL): 35.29 ± 4.08 PC: Oseltamivir IC_50_ (µg/mL): 5.88 ± 0.89 NC: Not reported Mode of action: - Inhibits viral growth; - Inhibits neuraminidase (NA) activity. (in vitro)	[31]
Brazil and China	95% Ethanol	Human immunodeficiency virus type 1 (HIV-1) Model: CD4^+^ lymphocytes and microglial cell cultures At 66.6 µg/mL give inhibition 85 and 98% PC: Zidovudine (AZT) or indinavir NC: Culture medium alone Mode of action: Inhibits viral entry. (in vitro)	[51]
Southern Brazil	Ethanol	Influenza A/PR/8/34 (H1N1) Assay: Plaque reduction assay Model: Mice Dose: 10 mg/kg Propolis (EC_50_ µg/mL): - AF-05: 62.0 ± 3.4 - AF-06: 60.0 ± 3.7 - AF-07: 59.1 ± 3.9 - AF-08: 22.6 ± 2.0 - AF-17: 19.5 ± 0.8 - AF-18: 45.2 ± 7.8 - AF-19: 34.6 ± 4.0 - AF-20: <10 - AF-M1: 60.3 ± 8.1 - AF-M2: 101.2 ± 2.0 - AF-M3: 111.6 ± 25.1 - AF-G1: 54.3 ± 3.5 - AF-G12: 36.8 ± 16.9 PC: Ribavirin EC_50_ = 20.2 ± 11.7 µg/mL NC: 1% ethanol solution Mode of action: Not reported. (in vitro and in vivo)	[52]
Cuautitlan Izcalli, State of Mexico	70% Ethanol	Pseudo Rabies Virus (PRV) Model: Monolayers of Madin-Darby bovine kidney (MDBK) cells Assay: Plaque assay Tested dose: 0.5 mg /mL PC: Not reported NC: Not infected MDBK cell culture Plaque forming unit: 8.7 Mode of action: - Possible damage to the viral envelope proteins; - Affects the penetration of the virus and its replication cycle. (in vitro)	[53]

NR: Not reported, NC: normal control, MDBK cell: Madin-Darby bovine kidney cells, PPE: Propolis extract ACF^®^, AF: propolis extract, PC: positive control.

**Table 2 foods-10-01776-t002:** List of isolated anti-viral compounds from propolis.

Origin	Compound Names	Antiviral Activities/Species	References
Southern Brazil	Melliferone	Human immunodeficiency virus (HIV) Model: H9 lymphocytes IC_50_: 0.205 µg/mL PC: AZT IC_50_: 500 µg/mL EC_50_: 0.00289 µg/mL NC: Infected cells with culture medium Mode of action: Inhibits viral replication. (in vitro)	[55]
	Moronic acid	Human immunodeficiency virus (HIV) Model: H9 lymphocytes IC_50_: 18.6 µg/mL EC_50_ < 0.1 µg/mL - PC: AZT IC_50_: 500 µg/mL EC_50_: 0.00289 µg/mL NC: Infected cells with culture medium Mode of action: Inhibits viral replication. (in vitro)	[55]
	Anwuweizonic acid	Human immunodeficiency virus (HIV) Model: H9 lymphocytes IC_50_: 2.14 µg/mL PC: AZT IC_50_: 500 µg/mL EC_50_: 0.00289 µg/mL NC: Infected cells with culture medium Mode of action: Inhibits viral replication. (in vitro)	[55]
	Betulonic acid	Human immunodeficiency virus (HIV) Model: H9 lymphocytes IC_50_: 1.8 µg/mL PC: AZT IC_50_: 500 µg/mL EC_50_: 0.00289 µg/mL NC: Infected cells with culture medium Mode of action: Inhibits viral replication (in vitro)	[55]
	4-Hydroxy-3-methoxypropiophenone	Human immunodeficiency virus (HIV) Model: H9 lymphocytes IC_50_: 18.8 µg/mL PC: AZT IC_50_: 500 µg/mL EC_50_: 0.00289 µg/mL NC: Infected cells with culture medium Mode of action: Inhibits viral replication. (in vitro)	[55]
	4-Hydroxy-3 methoxybenzaldehyde	Human immunodeficiency virus (HIV) Model: H9 lymphocytes IC_50_: >100 µg/mL PC: AZT IC_50_: 500 µg/mL EC_50_: 0.00289 µg/mL NC: Infected cells with culture medium Mode of action: Inhibits viral replication. (in vitro)	[55]
	3-(3,4-Dimethoxyphenyl)-2-propenal	Human immunodeficiency virus (HIV) Model: H9 lymphocytes IC_50_: 18.9 µg/mL PC: AZT IC_50_: 500 µg/mL EC_50_: 0.00289 µg/mL NC: Infected cells with culture medium Mode of action: Inhibits viral replication. (in vitro)	[55]
	12-Acetoxytremetone	Human immunodeficiency virus (HIV) Model: H9 lymphocytes IC_50_: 2.07 µg/mL PC: AZT IC_50_: 500 µg/mL EC_50_: 0.00289 µg/mL NC: Infected cells with culture medium Mode of action: Inhibits viral replication. (in vitro)	[55]
Uniflora Apicultores Associados (Olimpia, Brazil)/ Moravia, Czech Republic	Kaempferol	Human rhinovirus (HRV)-2, HRV-3, and HRV-4 Assay: Ulforhodamine B and real-time reverse transcription PCR Model: HeLa cells HRV-2: IC_50_ = 7.3 ± 4.54 µM PC: Ribavirin IC_50_ =270.1 ± 35.94 µM HRV-3: IC_50_ = 11.9 ± 0.42 µM PC: Ribavirin IC_50_ =307.9 ± 5.53.94 µM HRV-4 IC_50_ = 12.9 ± 1.15 µM PC: Ribavirin IC_50_ = 323.9 ± 31.16 µM NC: DMSO Mode of action: Blocks or reduces the entrance of the viruses into the cells to protect the cells from virus destruction and abate virus replication. (in vitro)	[13,15]
	*p*-Coumaric acid	Human rhinovirus (HRV)-2, HRV-3, and HRV-4 Assay: Ulforhodamine B and real-time reverse transcription PCR Model: HeLa cells HRV-2 IC_50_ = 371.2 ± 7.74 µM PC: Ribavirin IC_50_ = 270.1 ± 35.94 µM HRV-3 IC_50_ = 454.5 ± 3.16 µM PC: Ribavirin: IC_50_ = 307.9 ± 5.53.94 µM HRV-4 IC_50_ = 604.3 ± 50.93 µM PC: Ribavirin IC_50_ = 323.9 ± 31.16 µM NC: DMSO Mode of action: Blocks or reduces the entrance of the viruses into the cells to protect the cells from virus destruction and abate virus replication. (in vitro)	[13,15]
	Galangin	Human rhinovirus (HRV)-2, HRV-3, and HRV-4 Assay: Ulforhodamine B and real-time reverse transcription PCR Model: HeLa cells HRV-2 IC_50_ = 20.0 ± 8.07 µM PC: Ribavirin IC_50_ =270.1 ±35.94 µM HRV-3 IC_50_ = 116.2 ± 0.85 µM PC: Ribavirin IC_50_ = 307.9 ± 5.53.94 µM HRV-4 IC_50_ = 88.1 ± 28.71 µM PC: Ribavirin IC_50_ = 323.9 ± 31.16 µM NC: DMSO Mode of action: Blocks or reduces the entrance of the viruses into the cells to protect the cells from virus destruction and abate virus replication. (in vitro)	[13,15]
		Herpes simplex virus 1 (HSV-1) Assay: Plaque reduction Model: RC-37 cells IC_50_ (%): 0.00045; SI: 3.3 PC: Heparin-Na and acyclovir NC: Untreated cells Mode of action: Masks viral compounds which are necessary for adsorption or entry into host cells. (in vitro)	[13,15]
	Quercetin	Human rhinovirus (HRV)-2, HRV-3, and HRV-4 Assay: Ulforhodamine B and real-time reverse transcription PCR Model: HeLa cells HRV-2 IC_50_ = 34.1 ± 10.33 µM PC: Ribavirin IC_50_ =270.1 ± 35.94 µM HRV-3 IC_50_ = 15.5 ± 2.29 µM PC: Ribavirin: IC_50_ =307.9 ± 5.53.94 µM HRV-4 IC_50_ = 18.2 ± 4.14 µM PC: Ribavirin IC_50_ = 323.9 ± 31.16 µM NC: DMSO Mode of action: Blocks or reduces the entrance of the viruses into the cells to protect the cells from virus destruction and abate virus replication. (in vitro)	[13,15]
	Fisetin	Human rhinovirus (HRV)-2, HRV-3, and HRV-4 Assay: Ulforhodamine B and real-time reverse transcription PCR Model: HeLa cells HRV-2 IC_50_ = 40.9 ± 15.20 µM PC: Ribavirin IC_50_ =270.1 ± 35.94 µM HRV-3 IC_50_ = 67.1 ± 3.50 µM PC: Ribavirin IC_50_ =307.9 ± 5.53.94 µM HRV-4 IC_50_ = 66.4 ± 13.28 µM PC: Ribavirin IC_50_ = 323.9 ± 31.16 µM NC: DMSO Mode of action: Blocks or reduces the entrance of the viruses into the cells to protect the cells from virus destruction and abate virus replication. (in vitro)	[13,15]
	Chrysin	Human rhinovirus (HRV)-2, HRV-3, and HRV-4 Assay: Ulforhodamine B and real-time reverse transcription PCR Model: HeLa cells HRV-2 IC_50_ = 17.3 ± 9.83 µM PC: Ribavirin: IC_50_ =270.1 ± 35.94 µM HRV-3 IC_50_ = 16.1 ± 4.80 µM PC: Ribavirin IC_50_ =307.9 ± 5.53.94 µM HRV-4 IC_50_ = 24.4 ± 5.27 µM PC: Ribavirin IC_50_ = 323.9 ± 31.16 µM NC: DMSO Mode of action: Blocks or reduces the entrance of the viruses into the cells to protect the cells from virus destruction and abate virus replication. (in vitro)	[13,15]
		Herpes simplex virus 1 (HSV-1) Assay: Plaque reduction Model: RC-37 cells IC_50_ (%): 0.00003; SI: 20 PC: Heparin-Na and acyclovir NC: Untreated cells Mode of action: Masks viral compounds which are necessary for adsorption or entry into host cells. (in vitro)	[13,15]
	Luteolin	Human rhinovirus (HRV)-2, HRV-3, and HRV-4 Assay: Ulforhodamine B and real-time reverse transcription PCR Model: HeLa cells HRV-2 IC_50_ = 37.4 ± 2.10 µM PC: Ribavirin IC_50_ =270.1 ± 35.94 µM HRV-3 IC_50_ = 20.4 ± 2.63 µM PC: Ribavirin IC_50_ =307.9 ± 5.53.94 µM HRV-4 IC_50_ = 14.7 ± 7.86 µM PC: Ribavirin: IC_50_ = 323.9 ± 31.16 µM NC: DMSO Mode of action: Blocks or reduces the entrance of the viruses into the cells to protect the cells from virus destruction and abate virus replication. (in vitro)	[13,15]
	Acacetin	Human rhinovirus (HRV)-2, HRV-3, and HRV-4 Assay: Ulforhodamine B and real-time reverse transcription PCR Model: HeLa cells HRV-2: IC_50_ = 163.2 ± 18.97 µM PC: Ribavirin: IC_50_ = 270.1 ±35.94 µM HRV-3: IC_50_ = 107.6 ± 18.30 µM PC: Ribavirin: IC_50_ = 307.9 ± 5.53.94 µM HRV-4: IC_50_ = 102.3 ± 3.59 µM PC: Ribavirin: IC_50_ = 323.9 ± 31.16 µM NC: DMSO Mode of action: Blocks or reduces the entrance of the viruses into the cells to protect the cells from virus destruction and abate virus replication. (in vitro)	[13,15]
	Caffeic acid	Human rhinovirus (HRV)-2, HRV-3, and HRV-4 Assay: Ulforhodamine B and real-time reverse transcription PCR Model: HeLa cells HRV-2 IC_50_ = 67.2 ± 5.89 µM PC: Ribavirin IC_50_ =270.1 ± 35.94 µM HRV-3 IC_50_ = 52.2 ± 2.61 µM PC: Ribavirin IC_50_ =307.9 ± 5.53.94 µM HRV-4 IC_50_ = 66.1 ± 15.43 µM PC: Ribavirin IC_50_ = 323.9 ± 31.16 µM NC: DMSO Mode of action: Blocks or reduces the entrance of the viruses into the cells to protect the cells from virus destruction and abate virus replication. (in vitro)	[13,15]
	Ferulic acid	Human rhinovirus (HRV)-2, HRV-3, and HRV-4 Assay: Ulforhodamine B and real-time reverse transcription PCR Model: HeLa cells HRV-2 IC_50_ = 175.1 ± 29.10 µM PC: Ribavirin IC_50_ = 270.1 ±35.94 µM HRV-3: IC_50_ = 248.7 ± 22.30 µM PC: Ribavirin IC_50_ = 307.9 ± 5.53.94 µM HRV-4 IC_50_ = 232.3 ± 5.05 µM PC: Ribavirin IC_50_ = 323.9 ± 31.16 µM NC: DMSO Mode of action: Blocks or reduces the entrance of the viruses into the cells to protect the cells from virus destruction and abate virus replication. (in vitro)	[13,15]
Brazil	Apigenin	Anti-influenza virus Assay: Plaque reduction Model: MDCK A/PR/8/34(H1N1) EC_50_ = 15.3 ± 3.0 µg/mL A/Toyama/129/2011(H1N1) EC_50_ = 17.8 ± 8.7 µg/mL A/Toyama/26/2011(H1N1 EC_50_ = 8.1 ± 4.7 µg/mL PC: Ribavirin EC_50_ = 19.2 ± 7.5µg/mL NC: Distilled water Mode of action: Suppresses the stage of virus replication after adsorption and/or invasion. (in vitro)	[56]
	Artepillin C	Anti-influenza virus Assay: Plaque reduction Model: MDCK A/PR/8/34(H1N1) EC_50_ ˃ 40 µg/mL A/Toyama/129/2011(H1N1) EC_50_ ˃ 40 µg/mL A/Toyama/26/2011(H1N1 EC_50_ ˃ 40 µg/mL PC: Ribavirin EC_50_ = 19.2 ± 7.5µg/mL NC: Distilled water Mode of action: Suppresses the stage of virus replication after adsorption and/or invasion. (in vitro)	[56]
	Kaempferol	Anti-influenza virus Assay: Plaque reduction Model: MDCK A/PR/8/34(H1N1) EC_50_ = 38.2 ± 17.1 µg/mL A/Toyama/129/2011(H1N1) EC_50_ = 21.7 ± 5.5 µg/mL A/Toyama/26/2011(H1N1 EC_50_ = 24.8 ± 4.3 µg/mL PC: Ribavirin EC_50_ = 19.2 ± 7.5µg/mL NC: Distilled water Mode of action: Suppresses the stage of virus replication after adsorption and/or invasion. (in vitro)	[56]
	Caffeic acid	Anti-influenza virus Assay: Plaque reduction Model: MDCK A/PR/8/34(H1N1) EC_50_ >100 µg/mL A/Toyama/129/2011(H1N1) EC_50_ = 49.7 ± 5.0 µg/mL A/Toyama/26/2011(H1N1 EC_50_ > 100 µg/mL PC: Ribavirin EC_50_ = 19.2 ± 7.5µg/mL NC: Distilled water Mode of action: Suppresses the stage of virus replication after adsorption and/or invasion. (in vitro)	[56]
	Coumaric acid	Anti-influenza virus Assay: Plaque reduction Model: MDCK A/PR/8/34(H1N1) EC_50_ = 31.5 ± 1.3µg/mL A/Toyama/129/2011(H1N1) EC_50_ = 16.4 ± 6.6 µg/mL A/Toyama/26/2011(H1N1 EC_50_ = 27.0 ± 4.9 µg/mL PC: Ribavirin EC_50_ = 19.2 ± 7.5µg/mL NC: Distilled water Mode of action: Suppresses the stage of virus replication after adsorption and/or invasion. (in vitro)	[56]
	Caffeic acid phenethyl ester (CAPE)	Herpes simplex viruses (HSV-1 and HSV-2) Assay: Microscopic Fourier transform infrared spectroscopy (FTIR) Model: Mouse embryo fibroblasts (MEF) Tested doses: 10 and 50 µM PC: NR NC: Untreated cells (in vitro)	[57]

MDCK: Madin-Darby canine kidney cells, NR: Not reported, NC: normal control, PC: positive control.

**Table 3 foods-10-01776-t003:** The immunomodulatory effect of propolis.

Place of Propolis Collection	Type of Extract	Immunomodulatory Effect	References
Northeast of Algeria	85% Ethanol	Celiac Disease (immune-mediated enteropathy) Assay: Griess method, ELIZA, and immunofluorescence assay Model: Peripheral blood mononuclear cells (PBMCs) Tested doses: 1, 50, and 100 µg/mL Mode of action: - Increases NO and IFN-γ levels, and increases IL-10 levels; - Decreases iNOS expression and downregulates the activity of NFκB and pSTAT-3 transcription factors. (ex vivo)	[99]
Brazil	70% Ethanol	Antifungal immunity Assay: Flow-cytometry and ELISA Model: Human monocytes Tested doses: 5, 10, 25, 50, and 100 μg/mL Mode of action: - Upregulates TLR-4 and CD80; - Increases the fungicidal activity of monocytes. (in vitro)	[82]
Brazil	Ethanol	Antileishmaniasis Assay: Examination by SEM spectroscopy, Phagocytic Assay, ELISA Model: Human urine (in vitro), macrophages (in vivo) Tested doses: 5, 10, 25, 50, and 100 μg/mL (in vitro) 2.5, 5, or 10 mg/kg (in vivo) Mode of action: - Increases TNF-α and decreases IL-12 production; - Increases parasite interiorization by macrophages. (in vitro, in vivo)	[88]
Brazil	Propolis standard extract (dry extract)	Allergic asthma Assay: Flow cytometry, real time-PCR, and ELIZA Model: Bronchoalveolar lavage fluid (BALF) of allergic mice Tested doses: 150 mg/Kg every day for 17 days Mode of action: - Decreases pulmonary inflammation and mucus production as well as eosinophils and IL-5; - Enhances differentiation and frequency of lung MDSC and CD4^+^ Foxp3^+^ regulatory T cells. (in vitro, in vivo)	[70]
Northern Morocco	Ethanol	Immunomodulation Assay: Cytotoxic and cytostatic assays, MTT assay, and ELISA Model: MCF-7, HCT, THP-1, and PBMNCs cell lines IC_50_: 479.22, 108.88, and 50.54 μg/mL Tested doses: 125 and 250 μg/mL Mode of action: - Suppresses the TNF-α and IL-6 production in LPS-stimulated PBMNCs; - Increases IL-10 in a dose-dependent manner. (in vitro)	[100]
Brazil	0.1% Ethanol	Leishmaniasis Assay: Cytometric bead array assay, indirect immunofluorescence assay Model: PBMNCs Tested doses: 5 and 25 µg/mL Mode of action: Increases IL-4 and IL-17 and decreases IL-10. (in vitro)	[90]
Iran	30% Ethanol	Immunomodulation on tumor-bearing mice with disseminated candidiasis Assay: ELISA Model: Mouse mammary tumor Tested doses: 100 mg/kg Mode of action: Decreases IL-4 and IL-10 levels and increases TNF-α and IFN-γ levels. (in vitro)	[101]
Iran	Ethanol	Lipopolysaccharide-induced inflammation Assay: MTT assay, the Griess method, flow cytometry real-time PCR, and MTT assays Model: Murine macrophage (RAW 264.7) IC_50_: 15 ± 3.2 µg/mL Tested doses: 15, 1.5, 0.15 µg/mL Mode of action: Inhibits NO and ROS production and then decreases COX-2, IL-1β, and IL-6 gene expression. (in vitro)	[102]
Brazil	70% Ethanol	Immunomodulation Assay: MTT assay, ELISA, RT-qPCR, flow cytometry Model: Human DCs Tested doses: (5, 10, 20 and 40 μg/mL) Mode of action: Activates human DCs; induces the NF-kB signaling pathway and TNF-α, IL-6, and IL-10 production; inhibits the expression of hsamiR-148a and hsa-miR-148b; and increases of miR-155 expression. (in vitro)	[85]
Brazil	70% Ethanol	Immunomodulation Assay: ELISA, real-time PCR Model: Peritoneal macrophages and spleen cells in BALB/c mice Tested doses: 200 mg/kg, 0.1 mL for 3 consecutive days by gavage Mode of action: - Increases IL-1β production and TLR-2 and TLR-4 expression in peritoneal macrophages and spleen cells; - IL-6 production was also upregulated in the spleen cells. (In vivo)	[84]
Brazil Cuba Mexico	70% Ethanol	Immunomodulation Assay: MTT assay and ELISA Model: PBMNCS Tested doses: 0.2, 1.0, 2.0, 10.0, and 20.0 μg/mL Mode of action: Stimulates both TNF-α and IL-10 production by monocytes. (in vitro)	[103]
NR	Aqueous	Immunomodulatory activity in Zymosan-induced paw oedema Assay: AP complement assay, microtiter assay Model: Zymosan-induced paw oedema in mice strain ICR Tested doses: 150 mg/kg Mode of action: Inhibits the formation of edema by activation of alternative pathway (AP) complement. (in vivo)	[104]
Turkey	96% Ethanol	Immunomodulation Assay: MTT, high-pressure liquid chromatography, ELISA Model: PBMNCS Tested doses: NR Mode of action: Dose-dependent downregulation by induction of neopterin production and tryptophan degradation and inhibition of TNF-*α* and IFN-γ levels. (in vitro)	[105]
Brazil	70% Ethanol	Immunomodulation Assay: Cytotoxicity assay, ELISA Model: Peritoneal macrophages of BALB/c mice Tested doses: 5, 50, and 100 mg/well Mode of action: - Increases IL-1*β*; - Inhibits IL-10 and IL-6 production. (in vitro)	[106]
Brazil	70% Ethanol	Immunomodulation Assay: Radioimmunoassay, quantitative real time PCR, ELISA Model: C57BL/6 mice Tested doses: 200 mg/kg Mode of action: Increases the gene expression of TLR-2 and TLR-4. (in vivo)	[81]
Brazil	70% Ethanol	Ant-inflammatory Assay: 2,2-Diphenyl-1-picrylhydrazyl free radical (DPPH) scavenging method Griess reaction, MTT assay, ELISA Model: Raw 264.7 cells and a mouse leukemic monocyte macrophage cell line (ATCC TIB-71) Tested doses: 5, 10, 25, 50, and 100 μg/mL Mode of action: Inhibits NO production in macrophages and suppresses p38 MAPK, JNK1/2 pathways. (in vitro)	[107]
Brazil	Lyophilized samples	Immunomodulation Assay: Cotton Pellet Granuloma, ELISA, histopathology analysis Model: Swiss and Balb/c mice Tested doses: 5 mg/kg Mode of action: Decreases the concentration of TNF-α and IL-6 and increases TGF-β and IL-10. (in vivo)	[108]
Brazil	70% Ethanol	Antifungal immunity Assay: MTT, Flow Cytometry, ELISA Model: PBMNCS Tested doses: 5, 10, 25, 50, and 100 μg/mL Mode of action: - Downregulates the expression of TLR-2, HLA-DR, and CD80 and upregulates the expression of TLR-4; - High concentrations inhibited the production of TNF-α and IL-10. (in vitro)	[83]
Brazil	70% Ethanol	Immunomodulatory in acutely stressed mice Assay: ELISA Model: BALB/c male mice Tested doses: 200 mg/kg/day, 0.1 mL Mode of action: Increases IL-4 production in stressed mice. (in vivo)	[109]
Brazil	70% Ethanol	Immunomodulatory activity melanoma-bearing mice submitted to stress Assay: ELISA, Real-time PCR Model: C57BL/6 male mice Tested doses: 200 mg/kg Mode of action: Upregulates the expression of TLR-2, IL-10, and IFN-γ. (in vivo)	[110]
Brazil	70% Ethanol	Immunomodulation Assay: Real Time PCR, ELISA Model: BALB/c male mice Tested doses: 200 mg/kg/day, 0.1 mL Mode of action: Immunorestorative role in TLR-4 expression. (in vivo)	[111]
Brazilian propolis provided by Yamada Apiculture Center, Japan	Ethanol	Rheumatoid arthritis Assay: RT-PCR, ELISA, flow cytometry Model: DBA/1J mice Tested doses: (6.7 mg/g diet) and high dose (20 mg/g diet) propolis Mode of action: Inhibits production of IL-17 and the differentiation of Th17 cells. (in vivo)	[75]
Brazil	70% Ethanol	Fungicidal activity Assay: Flow cytometry, ELISA Model: PBMNCS Tested doses: 5, 10, 25, 50, and 100 μg/mL Mode of action: Downregulates the expression of TLR-2 and HLA-DR, inhibits TNF-α and IL-10 production, and increases fungicidal activity. (in vitro)	[86]
Brazilian propolis provided by Yamada Apiculture Center, Japan	Ethanol	Rheumatoid arthritis Assay: Cytotoxicity assays, Flowcytometry, Western blot analysis Model: splenocytes from Balb/c mice Tested doses: 12, 48 μg/mL Mode of action: - Inhibits IL-6 plus TGF-β-induced Th17 differentiation; - Suppresses IL-6-induced phosphorylation of STAT3. (in vitro)	[78]
Beekeeping Section, UNESP	70% Ethanol	Immunomodulation Assay: Real-time PCR, ELISA Model: Male BALB/c mice Tested doses: 200 mg/kg, 0.1 mL Mode of action: Inhibits the production of IFN-γ. (in vivo)	[112]
Bulgaria	Ethanol	Prophylactic activity against Gram-negative bacteria Assay: Negative Limulus amoebocyte lysate assay Model: Strain ICR mice Tested doses: 150 mg g-1 Mode of action: Production of Clq Macrophages, and change in the alternative complement pathway hemolysis. (in vivo)	[113]
Croatia	- Water-soluble derivative of propolis (WSDP) was prepared by freeze-drying ethanolic propolis extract - Ethanolic propolis extract was prepared by 80% (V/V) ethanol	Immunomodulatory effect against irinotecan-induced toxicity and genotoxicity Assay: Hematological analysis, peripheral blood micronucleus (MN) assay Model: Male albino mice of the Swiss strains Tested doses: 100 mg/kg Mode of action: Inhibits the growth of Ehrlich ascites tumors (EAT) by activation of macrophages and neutrophils, which inhibits Irinotecan induced toxicity. (in vivo)	[92]
Croatia	WSDP	Antimetastatic effect against lung cancer Assay: Flow cytometry, colorimetric Griess reaction Model: CBA inbred mice Tested doses: 50 or 150 mg/kg Mode of action: Suppression of metastasis by activation of macrophages and production of TNF-α, H_2_O_2_, and nitric oxide NO. (in vivo)	[114]
Brazil	70% Ethanol	Immunomodulation Assay: ELISA, spleen cells proliferation assay Model: Spleen cells of BALB/c male Tested doses: 2.5, 5, and 10 mg/kg, for 3 days Mode of action: Decreases splenocyte proliferation and stimulates IFN-γ production. (in vitro)	[115]
Brazil Croatia	WSDP	Antimetastatic effect against lung cancer Assay: Flow cytometry, Plaque-forming cells assay, hematological analysis Model: Male and female CBA inbred mice Tested doses: 50 or 150 mg/kg and 50 mg/mL Mode of action: Suppression of metastasis and increased level of LAF produced macrophages. (in vitro*,* in vivo)	[93]
NR	WSDP	Transplantable mammary carcinoma Assay: Colorimetric Griess reaction, hematological analysis Model: Male and female CBA inbred mice Tested doses: 50 or 150 µg/kg Mode of action: Inhibits tumor by macrophages activation, induces NO production, increases, the response of splenocytes to monoclonal antibodies, and inhibits [^3^ *H*]TdR incorporation into HeLa cells. (in vitro)	[116]
Brazil	70% Ethanol	Antileishmanial activity Assay: MTT assay, real-time PCR Model: PBMNCs Tested doses: 5, 10, 25 µg/mL Mode of action: Reduces the expression of CCL5 and IFN-γ. (in vitro)	[89]
Brazil	Hydroalcoholic extract	Antileishmanial activity Assay: Neutral red method Model: Male Balb/C albino mice (lineage of Mus musculus), Tested doses: 1.5 mg/kg/day Mode of action: NR (in vitro)	[87]
Brazil	70% Ethanol	Immunomodulatory activity in chronically stressed mice Assay: Griess Reaction, histopathological analysis Model: Male BALB/c mice Tested doses: 200 mg/kg Mode of action: Increases production of H_2_O_2_ and inhibits NO. (in vivo)	[117]
Bulgaria	WSDP	Immunomodulatory activity Assay: Popliteal lymph node assay Model: Macrophage Tested doses: 50 and 150 mg/kg Mode of action: Activation of macrophages, which induces the production of IL-1. (in vitro)	[118]
Croatia Brazil	WSDP	Immunomodulatory activity against Ehrlich ascites tumor Assay: Differential cell counts Model: Male albino mice of the Swiss strain Tested doses: 50 mg/kg Mode of action: Increases macrophage spreading activity. (in vivo)	[94]
Croatia	WSDP	Ehrlich ascites tumor Assay: Macrophage spreading assay, colorimetric Griess reaction, plaque-forming cells (PFC) assay, biuret method Model: Male and female CBA and Swiss albino inbred mice Tested doses: 50 mg/kg Mode of action: Increases cytotoxic T-cell, NK, and B cells activity. (in vivo)	[95]
Croatia Brazil	WSDP	Immunomodulatory activity in lung metastases Assay: Flow cytometry Model: Male and female CBA inbred mice Tested doses: 50 or 150 mg/kg Mode of action: Reduces metastases, delays tumor formation, and increases survival of treated animals. (in vivo)	[119]
Croatia Brazil	WSDP	Immunomodulatory activity in metastasis mammary carcinoma Assay: Griess method, plaque-forming cells assay, flow cytometry Model: Human cervical carcinoma cells (HeLa) Tested doses: 50 and 150 mg/kg Mode of action: Inhibits metastasis of mammary carcinoma, induces apoptosis, activates macrophages, increases CD4^+^ and CD8^+^ T cells, and induces the production of NO. (in vivo)	[96]
Bulgaria	70% Ethanol	(Parasitemia) *Trypanosoma cruzi* Assay: Flow cytometry Model: Swiss mice Tested doses: 25 to 100 mg/kg Mode of action: Preferential expansion of CD8^+^, inhibits CD4^+^ CD69^+^ and CD8^+^ CD69^+^ in CD4^+^ CD44^+^ and CD8^+^ CD44^+^, and decreases CD8^+^CD62L in *Trypanosoma cruzi*-infected mice. (in vivo)	[120]

ELISA: Enzyme-linked immunosorbent assay, PBMNCs: Peripheral blood mononuclear cells, NR: not reported, STAT3: signal transducer and activator of transcription 3, IL-10: Interleukin 10, IL-6: Interleukin 6, IL-1*β*: Interleukin IL-1*β*, TLR-4: Toll-like receptor-4, TLR-2: Toll-like receptor-2, TNF-α: Tumor necrosis factor alpha, MTT assay: (3-[4,5-Dimethylthiazol-2-yl]-2,5 diphenyl tetrazolium bromide) assay, TGF-β: Transforming growth factor-β, HLA-DR: Human leukocyte antigen—DR isotype.

**Table 4 foods-10-01776-t004:** List of clinical application of propolis as anti-viral and immunomodulatory agents.

Activity/Disease	Therapeutic Effect	Preclinical/Clinical Trials/ Number of Participants	Dose/Administration Route	References
Anti-viral/genital herpes (HSV-2)	Heal genital herpetic lesions and reduce local symptoms	Randomized controlled trials/90 p	Four times daily for 10 days/topical	[122]
Anti-viral/HSV-1	Treat herpetic skin lesions	NR	Propolis 3%/topical	[121]
Anti-viral/Herpes zoster	Heal skin lesion and reduce pain	Clinical trial/60 p	Propolis lotion (3 times/day topical) + Acyclovir (400 or 800 mg oral) for 28 days	[123,124]
Anti-viral/Herpes labialis	Reduce pain, short healing time with painless incrustation	Randomized, double-blind trial/150 p	Propolis (0.1%, 0.5% and 1%)/topical	[125]
Anti-viral/Herpes labialis	Reduce pain, burning, itching, tension, and swelling	Randomized, single-blind trial/379 p	Propolis 0.5%/5 times per 5 days/topical	[126]
Anti-viral/Herpes labialis	Treat episodes of herpes labialis	Randomized, controlled double-blind study/397 p	Propolis 0.5%/(0.2 g) 5 times per 5 days/topical	[127]
Anti-viral/dengue hemorrhagic fever virus	Reduce TNF-*α* levels and improve platelet counts	Randomized, double-blind, placebo-controlled trial/63 p	Propolis 200 mg three times a day for 7 days/oral	[3]

**Table 5 foods-10-01776-t005:** Estimated docking scores (in kcal/mol) and binding features for 40 anti-viral compounds, darunavir, and favipiravir against SARS-CoV-2 3C-like protease (3CL^pro^), papain-like protease (PL^pro^), RNA-dependent RNA polymerase (RdRp), helicase (NSP13) receptor-binding domain (RBD) of the spike protein, and human angiotensin-converting enzyme 2 (ACE2).

**No.**	Compound Name	3CL^pro^		PL^pro^		RdRp		NSP13		RBD		ACE2	
		Docking Score (kcal/mol)	Binding Features (Hydrogen Bond Length in Å)	Docking Score (kcal/mol)	Binding Features (Hydrogen Bond Length in Å)	Docking Score (kcal/mol)	Binding Features (Hydrogen Bond Length in Å)	Docking Score (kcal/mol)	Binding Features (Hydrogen Bond Length in Å)	Docking Score (kcal/mol)	Binding Features (Hydrogen Bond Length in Å)	Docking Score (kcal/mol)	Binding Features (Hydrogen Bond Length in Å)
	Darunavir	−8.2	GLU166 (1.94, 2.88 Å), LEU167 (1.96 Å)	−3.8	GLU196 (2.18, 2.30 Å), MET199 (2.11 Å)	−4.4	TYR619 (2.11 Å), ASP760 (1.83 Å), GLU811 (2.24, 2.26 Å)	−7.1	LEU411 (2.93 Å), THR412 (2.18 Å), GLY414 (2.89 Å), LEU416 (2.12 Å), ASN556 (2.11 Å), ARG559 (1.97 Å)	−3.3	GLU498 (1.89 Å), GLN507 (2.17 Å), SER508 (1.92 Å)	−8.5	TYR202 (1.97 Å), GLU208 (3.02 Å), LYS562 (2.81 Å)
	Favipiravir	−4.3	GLU166 (1.96, 2.26 Å), ARG188 (1.89 Å), THR190 (2.15 Å)	−4.0	GLU196 (1.99 Å)	−4.8	TRP617 (1.88 Å), ASP761 (1.96 Å), ALA762 (1.80 Å), TRP800 (2.20 Å)	−4.3	PRO513 (2.18 Å), ASN515 (1.89 Å), THR531 (1.76 Å), ASP533 (2.19 Å)	−4.2	LYS472 (2.05 Å), SER508 (2.07 Å), GLU498 (2.20 Å)	−4.4	ASP206 (2.07 Å), GLU208 (1.98 Å), ALA398 (1.80 Å)
1	Retusapurpurin A	−9.4	LEU141 (2.22 Å) GLY143 (2.93 Å), SER144 (1.90 Å), CYS145 (2.25 Å)	−8.0	ARG157 (2.17 Å), GLU196 (2.70 Å), MET201 (2.20 Å)	−7.5	LYS621 (2.42 Å), LYS798 (2.89 Å), TRP800 (2.29 Å), GLU811 (2.23 Å)	−9.4	ARG408 (1.88, 2.11 Å) ASN515 (1.99, 2.15 Å),	−7.2	TYR365 (1.71 Å), ALA366 (2.14 Å)	−10.4	ASP206 (2.23 Å), ASN210 (2.16, 2.24 Å)
2	Capillartemisin A	−8.9	HIS163 (1.99 Å), HIS164 (2.02 Å), THR190 (1.86 Å), GLN192 (2.32 Å)	−7.4	ASP157 (1.81 Å), ASN260 (1.94 Å)	−6.7	TYR619 (1.99 Å), ASP760 (2.08 Å), GLU811 (1.93 Å)	−7.0	ARG177 (2.37 Å), ASN178 (2.91Å), ASN515 (1.83, 1.98, 2.02 Å), THR531 (2.10 Å), HIS553 (2.40 Å),	−4.8	TYR463 (1.91 Å), GLU498 (2.33 Å), GLN507 (2.01, 2.22 Å)	−8.9	GLU208 (2.35 Å), ALA396 (1.84 Å), GLU564 (1.75 Å)
3	Artepillin C	−8.8	TYR54 (2.12, 2.34 Å), CYS44 (2.46 Å), GLU166 (2.25 Å), ASP187 (2.02 Å)	−8.0	ASP157 (1.87 Å), ASN260 (1.87 Å)	−7.0	ASP760 (2.09 Å), SER814 (1.89 Å)	−6.6	ARG408 (2.02 Å), LEU411 (2.84 Å), LEU416 (2.11 Å)	−4.4	TYR463 (2.08 Å)	−9.0	GLU208 (2.35 Å), GLU564 (1.78 Å)
4	(*E*)-3-[4−Hydroxy-3-(2-hydroxy-3-methylbut-3-en-l-yl)-5-(3-methybut-2-en-l-yl)phenyl] propenoic acid	−8.7	MET49 (2.01 Å), TYR54 (2.24 Å), GLU166 (1.81, 2.13 Å)	−7.9	ASP157 (1.89 Å), TYR257 (2.02 Å), ASN260 (1.91 Å)	−6.3	SER759 (2.99 Å), ASP760 (1.74, 1.88 Å), TRP800 (1.86 Å), GLU811 (2.08 Å)	−6.4	ARG408 (1.95Å), LEU411 (2.12 Å), LEU416 (1.72 Å)	−2.7	SER508 (1.92 Å)	−8.8	GLN98 (1.86 Å), GLU208 (2.19 Å), GLU564 (2.06 Å)
5	Baccharin	−8.7	TYR54 (2.06, 2.33 Å), CYS44 (2.19 Å), LEU141 (2.32 Å), GLY143 (2.82 Å), ASP187 (2.45 Å), GLN189 (1.90 Å)	−8.2	ASP157 (1.87 Å), ARG159 (2.01 Å), TYR261 (2.00 Å)	−6.8	ASP618 (2.19 Å), ASP760 (1.99 Å), ASP761 (2.15 Å), SER814 (2.07 Å)	−6.5	ARG177 (2.48 Å), ASN178 (2.80 Å), SER485 (2.01 Å), ASN515 (1.93 Å), THR531 (1.92 Å)	−5.5	TYR463 (1.89 Å), GLU498 (1.88 Å), SER508 (1.87 Å), LEU506 (2.13 Å), GLN507 (2.17, 2.60 Å)	−8.5	GLN98 (2.23 Å), GLU208 (3.07 Å), SER563 (2.05 Å), TRP566 (2.19 Å)
6	(*E*)-2,3-Dihydroconiferyl p-coumarate	−8.6	LEU141 (1.96 Å), GLY143 (3.06 Å), SER144 (2.14 Å), CYS145 (2.96 Å), THR190 (2.05 Å)	−7.6	LYS150 (1.97 Å), LEU155 (2.01 Å), ASP259 (1.94 Å)	−5.7	ASP760 (2.14 Å), CYS813 (2.63 Å), SER814 (1.96 Å)	−6.8	ASN515 (2.52 Å), ASP533 (2.07Å)	−3.0	GLU498 (1.83 Å), GLN507 (2.38 Å)	−8.0	LEU95 (1.91 Å), GLN98 (1.97 Å), ASN210 (1.94 Å), TRP566 (2.23 Å)
7	Quercetin	−8.5	HIS164 (2.23 Å), ASP187 (1.91 Å), THR190 (2.08, 2.08 Å), GLN192 (1.51 Å)	−6.8	LYS150 (2.36 Å), GLU160 (2.28 Å), ASN260 (1.79 Å), TYR266 (2.98 Å), THR294 (2.09 Å), ALA239 (3.01 Å)	−6.6	TYR619 (1.99, 2.03 Å), ASP761 (2.10 Å), GLU811 (2.33 Å)	−6.9	ASN178 (2.25 Å), LEU416 (1.94 Å)	−4.7	PHE361 (2.00, 2.08 Å), TYR365 (2.89 Å), TYR463 (2.16 Å)	−9.1	GLU208 (1.88 Å), ASN210 (2.06, 2.16, 2.85 Å), SER563 (2.00 Å), GLU564 (2.92 Å)
8	Sakuranetin	−8.5	HIS164 (2.22 Å), THR190 (2.01 Å)	−6.9	ASP295 (1.65 Å),	−6.7	TYR619 (1.86 Å), ASP761 (2.43 Å)	−6.9	PRO405 (1.84 Å), ASN515 (1.96 Å)	−4.9	SER363 (1.93 Å), TYR365 (1.77 Å), ASN364 (1.99, 2.73 Å)	−8.8	ASP206 (1.99 Å), ASN210 (1.87, 2.00 Å)
9	Kaempferol	−8.4	HIS164 (2.04 Å), ASP187 (1.99 Å), THR190 (1.81 Å)	−7.2	LEU255 (1.91 Å), ASN260 (1.96 Å), THR294 (1.83 Å)	−6.4	TRP617 (2.45 Å), TYR619 (1.84 Å), ASP761 (2.27 Å), TRP800 (2.12 Å)	−6.9	ASN178 (2.03 Å), LEU416 (1.89 Å), ASN556 (1.87 Å), ARG559 (2.70 Å)	−4.9	SER363 (2.08 Å), TYR365 (1.69 Å), ASN464 (1.92, 2.67 Å)	−8.9	ASP206 (1.86 Å), GLU208 (1.89 Å), ASN210 (2.02, 2.15, 2.83 Å) SER563 (2.02 Å), GLU564 (2.99 Å)
10	Isorhamnetin	−8.3	HIS164 (2.36 Å), ASP187 (1.98 Å), THR190 (2.07 Å)	−6.8	GU160 (1.99 Å), ASN260 (1.89 Å), TYR266 (2.16 Å), THR294 (1.84 Å)	−6.5	LYS621 (2.32 Å), CYS622 (2.19 Å), ASP760 (1.77 Å), ASP761 (1.90 Å), GLU811 (2.27 Å)	−6.6	ASN178 (2.15 Å), ARG408 (2.18 Å), PRO513 (1.83 Å), THR531 (1.76 Å), ASN515 (2.89 Å)	−4.9	SER363 (2.05 Å), TYR365 (1.75 Å), ASN464 (2.02, 2.63 Å)	−9.1	ASP206 (2.23 Å), GLU208 (1.83 Å), ASN210 (2.01, 2.09, 3.03 Å), LYS562 (2.25 Å), SER563 (1.95 Å), GLU564 (2.98 Å)
11	Sakuranetin- 4′-methylether	−8.2	HIS164 (2.25 Å), GLN192 (2.21 Å)	−7.0	ARG159 (1.93 Å), THR294 (1.87 Å)	−6.5	ASP761 (2.40 Å)	−7.0	ARG408 (2.13 Å), ASN515 (2.12, 2.30 Å)	−4.9	TYR365 (2.21 Å), ASN464 (2.22, 2.26 Å)	−8.8	ASN210 (1.96, 1.98 Å)
12	Pinobanksin- 3-acetate	−8.1	HIS164 (2.21 Å), GLU166 (1.96 Å), ASP187 (1.87 Å)	−6.8	TYR266 (2.92 Å), ASP295 (2.08 Å)	−6.0	LYS621 (2.45, 2.77 Å), CYS622 (1.89 Å), LYS798 (2.21 Å)	−6.7	ASN178 (2.22 Å), THR409 (2.00 Å), ARG559 (2.29 Å)	−5.4	PHE361 (1.71 Å), TYR365 (2.09 Å), ASN464 (2.14 Å)	−8.8	ASN210 (2.05, 2.16, 2.92 Å) SER563 (2.21 Å)
13	Calycosin	−8.1	ASP187 (2.0 Å), THR190 (2.15 Å)	−7.4	LYS150 (2.42 Å), GLU160 (2.29 Å), THR294 (1.92 Å)	−6.2	LYS619 (2.59 Å), LYS621 (2.22 Å), ASP623 (2.08 Å), LYS798 (2.33 Å)	-7.0	ASN176 (1.97, 2.90 Å), TYR197 (1.78 Å), ASP533 (1.84 Å)	-5.1	PHE361 (2.10 Å), TYR365 (1.65 Å)	−8.7	ASP206 (2.10 Å), ASN210 (1.73 Å), LYS562 (2.10 Å), SER563 (2.08 Å), GLU564 (3.04 Å)
14	Myricetin	−8.0	HIS164 (2.23 Å), ASP187 (1.91 Å), THR190 (2.02, 2.09 Å), GLN192 (2.52 Å)	−7.0	ARG159 (3.01, 3.02 Å), ALA239 (2.84 Å), TYR257 (2.79 Å), ASN260 (1.83 Å), TYR261 (2.63 Å), THR294 (2.04 Å)	−6.8	TYR619 (2.22 Å), LYS621 (2.83 Å), ASP760 (1.92 Å), ASP761 (1.95 Å), GLU811 (2.36 Å)	−7.0	ASN178 (2.09 Å), PRO405 (1.80 Å), LEU416 (1.85 Å), ASN556 (1.91 Å), ARG559 (2.77Å)	−4.8	ARG360 (2.95 Å), PHE361 (1.98, 2.32 Å), TYR363 (2.17 Å), ASN464 (1.84 Å)	−8.6	ASP206 (1.88 Å), ASN210 (1.99,2.16 ALA396 (1.82, 1.84 Å), SER563 (2.09 Å), GLU564 (3.02 Å)
15	Apigenin	−8.0	HIS164 (2.39 Å), ASP187 (1.96 Å), THR190 (1.89 Å)	−6.5	LYS150 (1.96 Å), TYR261 (2.71 Å), THR294 (1.97 Å)	−6.8	TRP617 (2.48 Å), TYR619 (1.86 Å), ASP761 (2.26 Å), TRP800 (2.10 Å)	−6.8	LEU416 (1.95 Å), ASN556 (1.86, 2.19 Å)	−5.3	SER363 (2.06 Å), TYR365 (1.70 Å), ASN464 (1.96, 2.65 Å)	−8.7	ASP206 (1.92 Å), ASN210 (1.83, 1.86 Å), SER563 (2.00 Å)
16	Daidzein	−8.0	HIS164 (3.09 Å), THR190 (2.04 Å)	−6.8	LEU155 (2.69 Å), THR294 (1.87 Å)	−5.9	ASP623 (1.68 Å), TYR619 (2.13 Å), ASP761 (1.71 Å)	−6.8	ALA406 (2.13Å), ASN515 (1.96Å)	−5.9	PHE361 (1.69 Å), TYR365 (2.10 Å), ASN364 (1.81 Å), LEU482 (2.07 Å)	−7.9	GLN198 (2.58 Å), GLU564 (1.91 Å)
17	Chrysin	−7.8	GLU166 (2.09 Å), THR190 (1.91 Å)	−6.9	ARG159 (1.79, 2.19, 2.94 Å), THR294 (1.84 Å)	−6.2	LYS621 (1.99 Å), ASP623 (1.91, 2.17 Å)	−7.1	ASN178 (1.96, 2.12 Å), ASP533 (2.07 Å)	−5.6	SER363 (2.04 Å), TYR365 (1.74 Å), ASN464 (2.03, 2.71 Å)	−8.5	ASN210 (2.03, 2.07 Å), SER563 (2.02 Å), GLU564 (3.03 Å)
18	Liquiritigenin	−7.8	ASP187 (1.95 Å), THR190 (1.96 Å)	−7.0	GLY156 (1.99 Å), ASP295 (1.84 Å)	−6.2	LYS621 (2.10 Å), CYS622 (2.95 Å), ASP623 (2.07 Å), ASP760 (2.35 Å)	−7.0	ASN178 (2.08 Å), LEU416 (2.31, 2.54 Å), ASP533 (2.26 Å)	−5.9	GLU420 (1.77 Å), TYR463 (1.98 Å), GLY510 (1.85 Å)	−8.4	ASP188 (1.92 Å), GLN192 (1.79 Å), SER545 (2.04 Å)
19	Luteolin	−7.8	TYR54 (2.76 Å), HIS164 (2.17 Å), THR190 (2.06 Å), GLN192 (2.42 Å)	−7.1	LYS150 (2.43 Å), GLU160 (2.29 Å), TYR266 (2.17 Å), THR294 (1.90, 2.04 Å)	−6.9	TYR619 (2.00, 2.01 Å), ASP761 (2.06 Å), GLU811 (2.33 Å)	−6.9	ASN78 (2.30 Å), LEU416 (2.09 Å), ASN556 (2.07Å)	−4.9	PHE361 (1.77, 1.96 Å), TYR365 (2.45 Å), TYR463 (2.16 Å)	−8.9	ASP206 (1.88, 1.98 Å) ASN210 (1.82, 1.86 Å) SER563 (2.02 Å)
20	Vestitone	−7.8	HIS163 (2.04 Å), GLU166 (2.11 Å), ASP187 (2.54 Å)	−7.5	ARG159 (1.94 Å), TYR261 (2.15, 2.44 Å)	−5.9	ASP623 (1.99 Å), ASP760 (2.13 Å)	−7.5	ASP533 (1.68 Å)	−4.7	TYR365 (1.87 Å), TYR463 (1.95 Å)	−7.6	LYS94 (1.74 Å), ALA396 (2.11 Å), TRP566 (2.28 Å)
21	Caffeic acid phenethyl ester (CAPE)	−7.8	LEU141 (2.11, 2.12 Å), GLY143 (2.44 Å), SER144 (2.15 Å), HIS163 (2.09 Å), GLU166 (2.35 Å)	−7.5	TYR266 (1.97 Å) ASP295 (1.78, 2.13 Å)	−6.5	TYR619 (2.13, 2.21 Å), ASP760 (1.96 Å), CYS813 (2.72 Å), SER814 (2.03 Å)	−6.5	THR415 (1.83, 2.09 Å)	−4.0	GLU498 (1.97, 2.04 Å)	−8.2	ASN210 (1.95, 2.47 Å) ALA396 (1.95, 1.97 Å), TRP564 (2.29 Å)
22	Formononetin	−7.7	PHE140 (2.66 Å), CYS145 (2.38 Å), GLU166 (1.97 Å)	−7.2	THR294 (1.77 Å)	−6.2	LYS621 (1.81 Å), LYS798 (1.94 Å)	−6.8	ASN515 (1.94 Å)	−5.3	PHE361 (1.69 Å), TYR365 (2.11 Å), ASN464 (1.81 Å)	−7.6	TYR196 (2.22 Å), GLU208 (1.70 Å), ASN210 (2.11 Å)
23	Pinocembrin	−7.7	GLU166 (2.12 Å), THR190 (1.76 Å)	−7.0	ARG159 (1.82, 2.10, 2.86 Å)	−6.0	TRP617 (3.08 Å), ASP761 (1.99 Å), ALA762 (1.81 Å), SER814 (2.35 Å)	−7.3	ASN178 (1.97, 2.15 Å), ASP533 (2.02 Å)	−5.1	SER363 (2.02 Å), TYR365 (1.79 Å), ASN464 (2.11, 2.71 Å)	−8.4	ASN210 (2.02, 2.14 Å) SER563 (2.11 Å), GLU564 (3.03 Å)
24	Isoliquiritigenin	−7.6	HIS164 (2.36 Å), ASP187 (1.98 Å), THR190 (2.07 Å)	−6.8	ALA239 (2.33 Å), ASP295 (1.77 Å)	−6.0	TYR617 (1.21 Å), ASP761 (2.18 Å), GLU811 (1.96 Å), SER814 (1.93 Å)	−7.1	ALA406 (2.09 Å), ARG408 (2.37, 3.08 Å), LEU416 (2.14, 2.23 Å), ASP533 (2.09 Å)	−4.3	PHE361 (1.95 Å), TYR365 (1.90 Å)	−8.1	ASN210 (1.96, 2.07 Å), SER563 (2.15 Å)
25	Violanone	−7.6	CYS44 (1.91 Å)	−7.3	LEU155 (2.04 Å), THR294 (2.02 Å)	−6.1	ASP164 (1.89 Å), LYS621 (2.31, 2.13 Å), TYR619 (2.30 Å), LYS798 (2.07 Å)	−7.3	ARG408 (2.12 Å), ASN515 (2.09 Å)	−4.9	SER363 (1.88, 1.96, 2.32 Å), ALA366 (2.13 Å), TYR463 (1.96 Å)	−8.1	LYS94 (2.08 Å), GLN98 (2.50 Å) ASN210 (2.20, 2.49 Å) ALA396 (1.90 Å), TRP566 (2.24 Å)
26	Drupanin	−7.5	HIS164 (1.96 Å), THR190 (1.83 Å), GLN192 (2.06 Å)	−7.0	ASP157 (1.87 Å), ASN260 (1.94 Å)	−6.3	ASP760 (1.85 Å)	−6.1	ASN178 (1.90 Å), TYR179 (1.72 Å), SER485 (2.08 Å), ASN515 (1.91 Å)	−4.0	GLU498 (1.68 Å), SER508 (2.01 Å)	−7.8	ASN210 (2.10 Å), GLU564 (1.70 Å)
27	Galangin	−7.5	HIS164 (2.21 Å), ASP187 (1.92 Å)	−6.8	ARG159 (1.85 Å), THR294 (1.73 Å)	−5.8	TRP617 (2.40 Å), ASP761 (2.33 Å), TRP800 (2.15 Å), SER814 (3.04 Å)	−6.6	LEU416 (1.91 Å), ASN556 (1.85 Å), ARG559 (2.71 Å)	−5.3	SER363 (2.06 Å), TYR365 (1.74 Å), ASN464 (2.01 Å)	−8.7	GLU208 (1.88 Å), ASN210 (2.04, 2.16, 2.86 Å), SER563 (1.99 Å), GLU564 (2.98 Å)
28	Pinobaskin	−7.5	HIS164 (2.23 Å), ASP187 (1.90 Å)	−6.7	ARG159 (1.83 Å), THR294 (1.73 Å)	−5.6	TRP617 (2.39 Å), ASP761 (2.30 Å), TRP800 (2.17 Å), SER814 (3.04 Å)	−6.3	LEU416 (1.91 Å), ASN556 (1.86 Å), ARG559 (2.70 Å)	−5.0	SER363 (2.04 Å), TYR365 (1.76 Å), ASN464 (2.01 Å)	−8.6	GLU208 (1.86 Å), ASN210 (2.04, 2.18, 2.85 Å), SER563 (1.99 Å), GLU564 (2.98 Å)
29	Pinostrobin	−7.4	SER144 (2.92 Å), HIS163 (1.98 Å)	−6.9	ARG159 (1.80 Å)	−6.1	ASP761 (2.23 Å)	−6.7	LEU416 (2.05, 2.30 Å), ARG559 (1.93 Å)	−5.1	TYR365 (2.20 Å), ASN464 (2.23, 2.27 Å)	−8.6	ASN210 (1.80, 1.98 Å)
30	Biochanin A	−7.4	THR190 (1.78 Å), GLN192 (3.06 Å)	−7.5	ARG159 (1.93 Å), TYR261 (2.13 Å) ALA239 (2.92 Å)	−6.3	TYR619 (3.05 Å), LYS621 (2.06 Å), ASP760 (1.82 Å), SER795 (2.79 Å)	−7.0	ASN178 (2.14 Å), ASN515 (1.96 Å)	−5.1	PHE361 (1.75 Å), TYR365 (2.11 Å), ASN464 (1.78 Å)	−7.9	TYR196 (2.22 Å), GLU208 (1.71, 2.01 Å), ASN210 (2.18 Å)
31	Pratensein	−7.4	LEU141 (2.17 Å), GLY143 (2.16 Å), ASP187 (2.85 Å)	−7.6	LEU155 (2.09 Å), THR294 (1.90 Å)	−6.9	LYS621 (2.13 Å), SER795 (2,11, 2.69 Å)	−7.2	ASN178 (2.16 Å), ASN515 (1.96 Å)	−4.9	GLU498 (1.83 Å), GLN507 (2.38 Å)	−7.8	GLN98 (2.72 Å), ASN210 (2.17 Å), GLU564 (1.85 Å)
32	2,2-Dimethyl chromene-6-propenoic acid	−7.3	THR190 (1.76 Å)	−7.3	ASP157 (2.07 Å)	−6.4	TRP617 (2.79 Å), ASP761 (1.88 Å)	−6.4	ASN178 (2.24 Å), THR531 (2.40 Å), ASN515 (2.03, 2.22 Å), ARG559 (2.89 Å)	−5.6	TYR467 (1.84 Å)	−7.2	GLU564 (1.95 Å)
33	Neovestitol	−7.2	GLU166 (2.03 Å)	−7.1	ARG159 (2.01 Å), TYR261 (2.11 Å)	−5.9	LYS798 (2.13 Å)	−6.8	ALA416 (2.16 Å), ARG408 (2.27 Å)	−5.5	SER508 (2.09 Å)	−7.4	ASN210 (1.80, 2.04 Å)
34	Rutin	−7.2	ASP187 (1.90 Å), HIS164 (2.25 Å), GLU166 (2.05 Å), GLN192 (2.86 Å), THR190 (2.19, 1.92 Å)	−5.2	GLU160 (1.95, 1.92 Å), SER163 (2.11 Å) GLU196 (2.12 Å), ARG176 (2.84 Å), MET201 (2.13 Å)	−4.5	ASP760 (2.23 Å)	−6.7	THR415 (1.84 Å), THR551 (2.51, 3.01 Å)	−3.3	SER363 (2.07 Å), TYR365 (1.80 Å), TYR463 (2.27 Å), ASN464 (1.98, 2.63, 3.05 Å)	−9.3	TYR196 (2.60 Å), ASP206 (2.01, 2.02 Å), GLU208 (2.20, 2.24 Å), ASN210 (2.33, 2.49 Å), TRP566 (2.32 Å)
35	Medicarpin	−6.5	GLY143 (2.18 Å), SER144 (2.62 Å), CYS145 (1.83 Å), HIS163 (1.99 Å),	−7.0	TYR266 (1.76 Å), ASP295 (1.82 Å)	−6.5	ASP761 (2.23 Å), HIS810 (2.17 Å), SER814 (2.02 Å)	−7.3	ARG408 (2.10 Å), THR415 (1.90 Å), LEU 416 (2.05 Å), ARG559 (1.89, 2.63 Å)	−5.6	ARG417 (2.14 Å), GLN507 (2.33 Å), SER508 (2.08 Å), GLY510 (1.97 Å)	−7.7	LYS94 (2.55 Å)
36	Ferulic acid	−5.9	CYS44 (2.48 Å), TYR54 (2.20, 2.44 Å), GLU166 (2.25 Å), ASP187 (2.01 Å)	−5.3	ASP157 (1.88 Å) ASN260 (2.13 Å)	−4.9	TRP617 (2.77 Å), ASP761 (1.83 Å)	−6.1	LYS201 (2.11, 2.79 Å) SER484 (1.87 Å), ASN515 (1.94 Å)	−5.0	GLU420 (1.62 Å), GLY510 (2.12 Å)	−6.1	ASN210 (1.76, 1.83 Å), ALA396 (1.88 Å)
37	*m*-Coumaric acid	−5.8	CYS44 (1.74 Å), TYR54 (2.01 Å), HIS164 (2.02 Å), ASP187 (1.82 Å)	−5.0	ARG159 (2.01 Å), ASN260 (1.75 Å), TYR261 (1.85 Å)	−5.5	ASP760 (1.95 Å), ASP761 (1.99 Å)	−6.8	LYS201 (2.11 Å), ASP482 (1.96 Å), SER485 (1.77 Å), ASN515 (1.92 Å)	−5.0	ASP481 (1.78 Å), GLU485 (1.83 Å)	−6.0	ASN210 (1.78 Å), ALA396 (1.86 Å), TRP566 (2.23 Å)
38	*p*-Coumaric acid	−5.8	CYS44 (1.73 Å), TYR54 (2.04 Å), HIS164 (2.03 Å), ASP187 (1.83 Å)	−5.2	ASP157 (1.99 Å), ASN260 (1.89 Å)	−4.9	ASP761 (1.81 Å), TRP800 (2.45 Å)	−5.3	ASN176 (2.88 Å), SER485 (2.06 Å)	−4.3	TYR467 (2.06 Å), SER508 (1.78Å), TYR519 (1.78 Å)	−5.5	ASN210 (2.05 Å), ALA396 (1.82 Å), TRP566 (2.22 Å)
39	Caffeic acid	−5.7	MET49 (1.95 Å), TYR54 (2.15 Å), GLU166 (2.05, 2.19 Å)	−4.9	ASN260 (1.79 Å), TYR266 (2.30 Å), THR294 (1.78, 1.90 Å)	−5.2	TRP617 (3.09 Å), ASP761 (1.73, 1.89 Å), HIS810 (2.07 Å), GLU811 (1.84 Å)	−6.5	LYS201 (2.11 Å), ASP482 (1.95 Å), SER485 (1.79 Å), ASN515 (1.81 Å)	−4.4	ARG417 (2.38 Å), GLU420 (1.72 Å), GLY510 (1.93, 2.04 Å)	−6.3	ASN210 (1.96, 1.99 Å), ALA396 (1.82 Å), TRP566 (2.24 Å)
40	*Trans*-Cinnamic acid	−5.5	CYS44 (1.74 Å), TYR54 (2.01 Å), ASP187 (1.82 Å)	−5.0	ARG159 (1.78, 2.10 Å), TYR261 (1.97 Å)	−5.1	TRP617 (2.73 Å), ASP761 (1.81 Å), TRP800 (2.42 Å)	−5.0	ASN176 (2.57 Å), SER485 (1.82 Å)	−4.7	TYR467 (1.87 Å), SER508 (1.81 Å)	−5.5	ASN210 (1.94 Å)

**Table 6 foods-10-01776-t006:** List of physicochemical, ADME, and toxicological properties of retusapurpurin A and baccharin.

Compound Name	MW ^a^	Log P ^b^	TPSA ^c^	nON ^d^	nOHNH ^e^	RBs ^f^	BBB+ ^g^	Caco2+ ^h^	HIA+ ^i^	AMES Toxicity	Carcinogenicity
Retusapurpurin A	522.6	6.2	98.4	7	2	5	0.41	74.84	98.97	Non toxic	Non carcinogenic
Baccharin	318.46	3.16	77.75	3	3	7	0.87	57.33	97.10	Non toxic	Non carcinogenic

^a^ Molecular weight ≤ 500 g/mol. ^b^ Partition coefficient logarithm between *n*-octanol and water ≤ 5. ^c^ Topological polar surface area ≤ 140 A^2^. ^d^ Number of hydrogen bond acceptors ≤ 10. ^e^ Number of hydrogen bond donors ≤ 5. ^f^ Number of rotatable bonds ≤ 10. ^g^ Blood–brain barrier ranged from −3 to 1.2. ^h^ Permeability < 25 (poor) and >500 (great). ^i^ %Human intestinal absorption < 25 (poor) and >80 (high).

## Data Availability

Not applicable.

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
