# Peer review of "Anti-Viral and Immunomodulatory Properties of Propolis: Chemical Diversity, Pharmacological Properties, Preclinical and Clinical Applications, and In Silico Potential against SARS-CoV-2"

_foods, 2021, doi:10.3390/foods10081776_

Round 1
Reviewer 1 Report
The review work concerns antivirial and immunomodulatory properties of propolis of different origin. An additional element of this work is “in silico” study against SARS-CoC-2. The topic is novel, and data well presented. The findings imply a new knowledge. However, the paper contains a large number of linguistic errors that often made it difficult to understand the content of individual phrases. Some examples of this type of errors are listed below. So, the manuscript needs a thorough linguistic improvement.
Page 3 line 121 “The Egyptians considered” should be changed as “ The Egyptians are considered”
Page 3 line 134 “Green propolis also has” should be changed as “Green propolis has also”
Page 3 line 148 “sugars” should be changed as “saccharides”
Page 4 line 160 “Figure 2” should be changed as “Figures 1-2”, since the Authors have mentioned phenolic acids and flavonoids together
Page 4 line 185 “HPLC analysis for” should be changed as “HPLC analysis of”
Page 4 line 190 “pinobaskin” should be changed as “pinobanksin”, the signature of chemical structure of this compound on Fig. 1 is also incorrect
Page 8 line 238 “ethanol” should be changed as “ethanolic”, see also at line 243
Page 8 line 248 “propolis collected from Canada” should be changed as “propolis originating from Canada”, “activity performed on” should be changed as “ activity was performed on”, by the way, what kind of activity was tested?
Page 8 line 257 “African region consider” should be changed as “African region is considered as the”
Page 8 line 262 “collected” should be changed as “originating”; “investigated” should be changed as “are investigated”
Page 8 line 263 “Propolis extracts tested” should be changed as “Propolis extracts were tested”
Page 8 line 268 “nwuweizonic acid” – this name is incorrect
Page 8 line 275 “Suppression effect detected when propolis extract added” should be chamged as “Suppresion effect was detected when propolis extract was added”
Page 8 line 282 “extracts with two different methods” should be changed as “extracts obtained with using two different methods”
Page 8 line 285 “propolis has highly activity against” should be changed as “ resulted on a higher activity of the extracts when compared with”
Page 9 line 299 “study conducted” should be changed as “study was conducted”
Page 9 line 300 “study carried” should be changed as “study was carried”
Page 9 line 302 “that infected” should be changed as “that were infected”
Page 9 line 306 It is not clear to whom these extracts were administered, what did the Authors mean writing “ 10 mg/kg is the most significant one”
Page 9 line 316 “compounds determined” should be changed as “the compounds were determined”
Page 9 line 318 “reduction assays used” should be changed as “reduction assays were used”
Page 10 “Flavinoids” should be changed as “Flavonoids” ( signature of the structure)
Page 22 line 352 According to the Authors five flavonoids have been evaluated, however, in the next sentence they mentioned only four of them (i.e., acacetin, galangin, chrysin and kaempferol)
Page 23 line 414 “colleagues” should be changed as “co-authors”
Page 24 line 481 “NO levels” should be changed as “NO (nitric oxide) levels”
Page 25 line 492 “water-soluble derivative” or “water-soluble extract” – which is correct?
Page 28 With respect to saline, what does it common with propolis, moreover, “Immunomodulatiry activitu” should be changed as “Immunomodulatory activity”
Page 33 line 541 “10th” should be changed as “10th day”
Page 33 line 543 “patients who healed with” should be changed as “patients treated with”,” more than healed with” should be changed as “more than those treated with”
Author Response
Reviewer 1
Comments and Suggestions for Authors
- The review work concerns antivirial and immunomodulatory properties of propolis of different origin. An additional element of this work is “in silico” study against SARS-CoC-2. The topic is novel, and data well presented. The findings imply a new knowledge. However, the paper contains a large number of linguistic errors that often made it difficult to understand the content of individual phrases. Some examples of this type of errors are listed below. So, the manuscript needs a thorough linguistic improvement.
Response: We would like to thank the reviewer for taking the time and effort necessary to provide such insightful comment; the manuscript has been carefully revised by native English speaker
- Page 3 line 121 “The Egyptians considered” should be changed as “ The Egyptians are considered”
Response: "are" is added
- Page 3 line 134 “Green propolis also has” should be changed as “Green propolis has also”
Response: Adjusted
- Page 3 line 148 “sugars” should be changed as “saccharides”
Response: "sugars" replaced with "saccharides"
- Page 4 line 160 “Figure 2” should be changed as “Figures 1-2”, since the Authors have mentioned phenolic acids and flavonoids together
Response: "Figure 2" replaced with "Figures 1-2"
- Page 4 line 185 “HPLC analysis for” should be changed as “HPLC analysis of”
Response: "for" replaced with "of"
- Page 4 line 190 “pinobaskin” should be changed as “pinobanksin”, the signature of chemical structure of this compound on Fig. 1 is also incorrect
Response: we would like to thank the referee for the comment but this is a pinobaskin compound (our compound) which is different from the pinobanksin compound; in accordance to the reference (Farooqui, T. and Farooqui, A.A., 2012. Beneficial effects of propolis on human health and neurological diseases. Front Biosci (Elite Ed), 4, pp.779-793.)
Reference:
- Farooqui, Akhlaq, F. Beneficial effects of propolis on human health and neurological diseases Tahira. Biosci. 2012, 4, 779–793.
- Page 8 line 238 “ethanol” should be changed as “ethanolic”, see also at line 243
Response: “ethanol” adjusted to “ethanolic”
- Page 8 line 248 “propolis collected from Canada” should be changed as “propolis originating from Canada”,
Response: "collected" replaced with "originating"
- “activity performed on” should be changed as “activity was performed on”, by the way, what kind of activity was tested?
Response: "was" is added/and "antiviral" activity is added
- Page 8 line 257 “African region consider” should be changed as “African region is considered as the”
Response: " is considered as the " is added
- Page 8 line 262 “collected” should be changed as “originating”; “investigated” should be changed as “are investigated”'
Response: Adjusted
- Page 8 line 263 “Propolis extracts tested” should be changed as “Propolis extracts were tested”
Response: "were" is added
- Page 8 line 268 “nwuweizonic acid” – this name is incorrect
Response: “nwuweizonic acid” is corrected to "anwuweizonic acid"
- Page 8 line 275 “Suppression effect detected when propolis extract added” should be changed as “Suppression effect was detected when propolis extract was added”
Response: "was" is added
- Page 8 line 282 “extracts with two different methods” should be changed as “extracts obtained with using two different methods”
Response: "were obtained using" is added
- Page 8 line 285 “propolis has highly activity against” should be changed as “ resulted on a higher activity of the extracts when compared with”
Response: Adjusted
- Page 9 line 299 “study conducted” should be changed as “study was conducted”
Response: "was" is added
- Page 9 line 300 “study carried” should be changed as “study was carried”
Response: "was" is added
- Page 9 line 302 “that infected” should be changed as “that were infected”
Response: "were" is added
- Page 9 line 306 It is not clear to whom these extracts were administered, what did the Authors mean writing “10 mg/kg is the most significant one”
Response: Four extracts were administered orally to mice with different doses (3 times daily/7 days); At 10 mg of propolis per kg of mice display the most significant activity”
- Page 9 line 316 “compounds determined” should be changed as “the compounds were determined”
Response: "were" is added
- Page 9 line 318 “reduction assays used” should be changed as “reduction assays were used”
Response: "were" is added
- Page 10 “Flavinoids” should be changed as “Flavonoids” ( signature of the structure)
Response: “Flavinoids” is replaced with “Flavonoids”
- Page 22 line 352 According to the Authors five flavonoids have been evaluated, however, in the next sentence they mentioned only four of them (i.e., acacetin, galangin, chrysin and kaempferol)
Response: We agree with the referee comment; "and quercetin were active against infectivity and replication only at high concentrations [17]. " is inserted
- Page 23 line 414 “colleagues” should be changed as “co-authors”
Response: Adjusted
- Page 24 line 481 “NO levels” should be changed as “NO (nitric oxide) levels”
Response: "(nitric oxide)" is added
- Page 25 line 492 “water-soluble derivative” or “water-soluble extract” – which is correct?
Response: "derivative" as mentioned in the original reference
- Page 28 With respect to saline, what does it common with propolis, moreover, “Immunomodulatiry activitu” should be changed as “Immunomodulatory activity”
Response:
-"saline" is changed to "aqueous"
- changed to “Immunomodulatory activity”
- Page 33 line 541 “10th” should be changed as “10thday”
Response: "days" is added
- Page 33 line 543 “patients who healed with” should be changed as “patients treated with”,” more than healed with” should be changed as “more than those treated with”
Response: Adjusted

Reviewer 2 Report
Detail comments see attached.

Author Response
The world is now in the midst of a Corvid-19 pandemic. It is a crisis for humankind. Propolis is expanding its use as a health food (supplement) and beverage. It is taken for the purpose of disease prevention and treatment with the expectation of antibacterial, antiviral, antiinflammatory, antitumor effects, etc
This review describes the antiviral and immunomodulatory properties of propolis and introduces the chemical diversity, pharmacological,preclinical, clinical applications and in silico studies for SARS-CoV-2 that causes Corvid-19. The authors are researching a lot of literature. This review is very timely and excellent. However, although it may have been written in a very hurry, there are insufficient expressions and English. The reviewers pointed them outbelow, but he wants the authors themselves to re-examine them.
Response: We would like to thank the reviewer for the nice comment and we agree with the reviewer and hence addressed these issues
[Compound name]
In general:
- Write the symbols E, Z, R, S indicating stereochemistry in italics.
Response: Adjusted
Uppercase and lowercase:
- In Figures 2 and 3, Table 2, the list (from line 61 to line 77), and
some parts of the text, uppercase letters are used for the initials of
compound names, but some are mistakenly used lowercase letters as below.
Response: Thank you very much for your helpful comments. We have amended the figures and table 2 accordingly.
- Line 73: p-coumaric acid
Response: Modified
-Line 165: (E)-2,3-dihydroconiferyl p-coumarate,
Response: Adjusted- Figure 2: 2,2-dimethylchromene-6-propanoic acid, trans-cinnnamic acid, pcoumaric acid, and m-coumaric acid.
-Figure 3: (E)-2,3-dihydroconiferyl p-coumarate
[Structural formulas of some compounds]
- Figures 1 and 2: Pinostobin, Pinobanksin-3-acetate, Formononetin, Vistitone, Biochanin A: It is not necessary to write the H atom attached to the aromatic ring, so erase it with the bond.
- Figure 2: The prenyl group (3-methyl 2-butenyl group) of Artepillin C, Capillartemisin, and Baccharin should be written as that of Drapanin. Figure 3: In (E)-2,3-dihydroconiferyl p-coumarate, it is not necessary to write the two H atoms attached to the two aromatic rings, so delete them.
Response: We thank the reviewer for the helpful comments. We have revised the figures accordingly; please check the new version
[1. Introduction]
- Page 2, lines 103-108: “In general, propolis consist of 50% resins, 30% wax, 10% essential oils, 5% pollen and 2% mineral salts and the main constituents of propolis are polyphenols, mainly flavonoids, phenolic acids, their esters and cinnamic acids [21], including several numbers of antiviral and immunomodulatory bioactive constituents i.e., kaempferol, ---[15,22,23].” According to other literature*, it was described that raw propolis is typically composed of 50% plant resins, 30% waxes, 10% essential and aromatic oils, 5% pollens and 5% other organic substances. Furthermore, terpenes (terpenoids) are also a class of components of propolis with biological activities. The reviewer would like to point out that "the main constituents of propolis" should read "a group of main compounds contained in propolis and having biological activity or pharmacological / food functionality".
*Nada Zabaioua and others: Biological properties of propolis extracts: Something new from an ancient product. Chemistry and Physics of Lipids, Volume 207, Part B, October 2017, Pages 214-222
Response: Adjusted
“In general, propolis consist of 50% resins, 30% wax, 10% essential oils, 5% pollen and 2% mineral salts and a group of main compounds contained in propolis and having biological activity or pharmacological / food functionality are polyphenols, mainly flavonoids, phenolic acids, their esters and cinnamic acids [21,22], including several numbers of antiviral and immunomodulatory bioactive constituents i.e., kaempferol, p-coumaric acid, apigenin, artepillin C, caffeic acid and caffeic acid phenyl ester [15,23,24].”
Page 2, line 102: “influenza viruses A/HlNl, NH3N2 [16],”This should read “influenza viruses A/HlNl and A/NH3N2 [16],” as an English correction. But, actually, according to the literature 16,
Response: “influenza viruses A/HlNl, and A/NH3N2 [1],”
- Page 2, line 104: “re” should read “are”.
Response: Adjusted
- Page 3, lines 109-110: “ --- to highlight the role of propolis and/or their derivatives as a ---”: This should read “ --- to highlight the role of propolis, its constituents, and/or their derivatives as a ---””
Response: Changed to “The main aim of this review is to highlight the role of propolis, its constituents, and/or their derivatives”
- [3. Chemical composition of propolis] (Page 3, line 142-)
- Page 3, lines 144-146: “ --- that propolis rich in polyphenolic compounds mainly flavonoids and cinnamic acids in which there are more than ---.” This should read “ --- that propolis is rich in polyphenolic compounds mainly containing flavonoids and cinnamic acids in which there are more than ---.”
Response: Adjusted to “Propolis is rich in polyphenolic compounds mainly containing flavonoids, cinnamic acids and esters(Figures 1-3).”
- Page 3, line 152: “(2S)-5-hydroxy-7-methoxyflavanone, (pinostrobin)”:
Delete this comma.
Response: Adjusted- Pages 3-4, lines 158-160: “7 flavonoids and other cinnamic and phenolic
compounds including caffeic acid, p-coumaric acid, ferulic acid, mcoumaric acid, trans-cinnamic acid, pinocembrin, and chrysin (Figure 2)” This should read “2 flavonoids (pinocembrin and chrysin) (Figure 1), trans-cinnamic acid and 4 phenolic cinnamic acid (caffeic acid, pcoumaric acid, ferulic acid, and m-coumaric acid) (Figure 2)”.
Response: Adjusted
- Page 4, line 161: “ --- in 2 propolis samples are different in apiary geographical locations in Italy ---”
This should read “ --- in 2 propolis samples which are different in apiary geographical locations in Italy ---”. [Add “which” between “samples” and “are”.]
Response: Modified to “besides many of volatile compounds had been identified in 2 propolis samples and are different in apiary geographical locations in Italy “
- Page 4, lines 164-166: “ --- including prenylated phenylpropanoids(drupanin, capillartemisin A, (E)-2,3-dihydroconiferyl p-coumarate, 2,2-dimethylchromene-6-propenoic acid, artepillin C and baccharin) (Figure3);”(E)-2,3-dihydroconiferyl p coumarate is not a prenylatedThese compounds are shown in Figure 2, not in Figure 3.
Response: Adjusted
- Page 4, line 169 and Page 7, Figure 3: “ -2-en-1yl)phenyl]” should read “ -2-en-1-yl)phenyl]”
Response: Adjusted
- Page 4, p.170-175: “Saito et al. evaluated the supercritical andconventional extraction procedures utilizing ethanol, methanol, water,hexane as solvents for 2 different types of propolis; green and red andfound that methanolic extraction are the best for both green and redpropolis contents yielding but supercritical CO2 fractionation of ethanolextract is much higher flavonoid content than crude ethanol extract[39].”
The reviewer tried to revise this part as follows, according to the literature [39]. “ Saito et al. evaluated conventional (Soxhlet) extraction with ethanol, methanol, water, and hexane as solvents and supercritical CO2 extraction for 2 different types of (green and red) propolis from Brazil, and found that methanolic extraction was the best method in terms of yield for both propolis, while fractionation of ethanolic extract of the red propolis with supercritical CO2 produced fractionated extracts with much higher flavonoid contents than those found in the original ethanolic extract.
Response: The authors completely agree with the referee. Accordingly, we modified and rearranged this paragraph.
“Saito et al. evaluated conventional (Soxhlet) extraction with ethanol, methanol, water, and hexane as solvents and supercritical CO2 extraction for 2 different types of (green and red) propolis from Brazil, and found that methanolic extraction was the best method in terms of yield for both propolis, while fractionation of ethanolic extract of the red propolis with supercritical CO2 produced fractionated extracts with much higher flavonoid contents than those found in the original ethanolic extract [2]”
- Page 4, p.176-179: “ ---, 13 of 38 compounds i.e., liquiritigenin, calycosin, calycosin (isomer), luteolin, isoliquiritigenin, formononetin, (3S)-vestitol, (3S)-neovestitol, retusapurpurin A, medicarpin, retusapurpurin A (isomer) and biochanin A had been tentatively identified---.” Although it is written as 13 of 38 compounds, the number of compound names listed is 12, which is one shortage.
Response: “12 of 38 compounds i.e., liquiritigenin, calycosin, calycosin (isomer), luteolin, isoliquiritigenin, formononetin, (3S)-vestitol, (3S)-neovestitol, retusapurpurin A, medicarpin, retusapurpurin A (isomer), and biochanin A had been tentatively identified in red propolis sample of Brazilian geographical source via HPLC-MS, and TLC-bioautography (Figure 1).”
- Page 4, line 180: “(3S)-vestitol” should read “(3S)-Vestitol”.
Response: Adjusted
- Page 4, line 182-183: “5 flavonoids compounds” should read “ 5 flavonoid compounds”.
Response: Adjusted
- Page 4, line 190: “pinobaskin” should read “pinobanskin”.
Response: We would like to thank the referee for the comment but this is a pinobaskin compound (our compound) which is different from the pinobanksin compound; in accordance to the reference (Farooqui, T. and Farooqui, A.A., 2012. Beneficial effects of propolis on human health and neurological diseases. Front Biosci (Elite Ed), 4, pp.779-793.)
Farooqui, Akhlaq, F. Beneficial effects of propolis on human health and neurological diseases Tahira. Front. Biosci. 2012, 4, 779–793.
- Page 4, line 197: “rutin, quercetin” should read “rutin, and
quercetin”
Response: Adjusted
- Page 6, Figure 1, title: Common flavonoids compounds should read Common
flavonoids compounds
Response: “Figure 1. Common flavonoid compounds isolated and identified from propolis of Apis mellifera L.”
- Page 7, Figure 2, title: “Common cinnamic acids compounds” should read
“Common cinnamic acid compounds”
Response: “Figure 2. Common cinnamic acid compounds isolated and identified from propolis of Apis mellifera L.“
- Page 7, Figure 3: This title is “Common polyphenol compounds ----“, but“(E)-3-[4-Hydroxy-3-(2-hydroxy-3-methylbut-3-en-1-yl)-5-(3-methylbut-2-en-1-yl)phenyl] propenoic acid” is not polyphenol compounds. It has onlyone phenolic hydroxyl group.
Response: Adjusted
[4. Anti-viral activity]
- Page 7, lines 216-217: “Among these, certain natural products from traditional medicine including propolis.” Revise the English of this sentence.
Response: This paragraph is rephrased to “Propolis has long been used to treat various diseases in particular viral infections and more recently COVID-19 [3]. Based on the unavailability of treatments, therapies or vaccines and emergence of drug resistant mutants against many of viruses, the exploring of novel candidates or compounds with different anti-viral mechanisms from safe and effective naturally sources are urgently needed. “
- Page 7, lines 220-222: “ --- while HSV-2 is mainly infecting the genital mucosa and consider is a sexually transmitted diseases, the two types of the viruses become drug (Acyclovir) resistance [43].”
This should read “ --- while HSV-2 is mainly infecting the genital mucosa and consider to be a sexually transmitted diseases, and the two types of the viruses become drug (Acyclovir) resistance [43].”
Response: Adjusted
- Page 7, line 225: “interestingly,” should read “Interstingly,”
Response: Adjusted
- Page 7, line 225-226: “ --- implies stronger activity than acyclovir
separately”.
This should read “ --- implies stronger activity than that of
acyclovir separately”.
Response: Changed to “implies stronger activity than that of acyclovir separately”.
- Page 7, line 229: “Brazilin extract” should read “the Brazilian
extract”.
Response: Adjusted
- Page 8, line 238: “Apis mellifera”: Write in italics.
Response: “Apis mellifera L.”
- Page 8, line 239: “show” should read “showed”.
Response: Adjusted
- Page 8, line 243: “investigated” should read “were investigated”.
Response: Adjusted
- Page 8, lines 244-247: “The two extracts of propolis had been add at different time of during the viral infection cycle caused considerable suppression of HSV-2 multiplication. Also, masking viral compounds that are responsible for entry or adsorption into host cells [47].” Correct the English of these two sentences.
Response: “Once the two propolis extracts were injected at different intervals during the viral infection cycle, HSV-2 proliferation was significantly reduced. Masking viral molecules responsible for entrance or adsorption into host cells is a possibility [4].”
- Page 8, line 248: “Another study conducted on propolis collected from
” should read “Another study was conducted on propolis collected
from Canada.”
Response: “Another study conducted on propolis originating from Canada”
- Page 8, line 251: “in vitro study ---” should read “In vitro study ---”.
Response: “In vitro”
- Page 8, line 251: “the France propolis” should read “the French
propolis”.
Response: “the French propolis”.
- Page 8, line 255: “On the other hand, HIV (Human Immunodeficiency
Virus), HIV causes the ---” should read “On the other hand, Human
Immunodeficiency Virus (HIV) causes the ---”.
Response: Changed to ““On the other hand, Human Immunodeficiency Virus (HIV) causes the ---”.
- Page 8, lines 269-271: Four compound names: Change the initials of the
four compound names to lowercase.
Response: “Additionally, eight compounds, four triterpenoids (melliferone, new compound; moronic acid; anwuweizonic acid, and betulonic acid) and four aromatic compounds (4-hydroxy-3-methoxypropiophenone, 4-hydroxy-3-methoxybenzaldehyde, 3-(3,4-dimethoxyphenyl)-2-propenal, and 12-acetoxytremetone) were isolated from Brazilian propolis. “ - Page 8, line 271: “These compounds tested” should read “These
compounds were tested”
Response: “These compounds were tested as anti-AIDS candidates using H9 lymphocytes as a model (Table 2) [5].”
- Page 8, line 274: “64 µg/mL.” Delete this period.
Response: Adjusted - Page 8, line 275: “when propolis extract added --” should read “when
propolis extract was added --”.
Response: Changed to ““when propolis extract was added”. - Page 8, lines 278-280: “Although acyclovir (positive control) showed a
good effect against viral DNA polymerase during the replication period,
but when combining with propolis extract giving more potent activity
against Varicella zoster virus (VZV) [52].”
Revise this English.
Response: “Although acyclovir (positive control) had a good effect on viral DNA polymerase during replication, when combined with propolis extract, it had a more potent effect on VZV [6].”
- Page 8, lines 281-283: “In another work a comparison between antiviral
effects of two types of Brazilian propolis (green/red) extracts with two
different methods (ultrasonic extraction and maceration).”
Revise this English.
Response: “A comparison of antiviral properties of two types of Brazilian propolis (green/red) extracts using two distinct procedures (ultrasonic extraction and maceration) was published in another study.”
- Page 8, line 284: “ultrasonic extraction” should read “ultrasonic
extract”
Response: Adjusted to ““ultrasonic extract”
- Page 8, line 285: “highly” should read “higher”.
Response: Adjusted
- Page 9, line 286: “red propolis more active than green one” should read
“red propolis are more active than green one.
Response: Changed
- Page 9, line 295: “ --- HRV-4, respectively while, p-coumaric acid shows
---” should read “ --- HRV-4, respectively, while p-coumaric acid shows
---”.
Response: Corrected
- Pge 9, line 304: “13 south Brazilian ethanolic extracts” should read
“13 ethanolic extracts of south Brazilian propolis
Response: Adjusted
- Page 9, lines 305-308: “So, the four extracts were tested as antiviralin vivo. Four extracts were administered orally with different doses (3times daily/7 days) following infection, 10 mg/kg is the most significantone [55]. ”This reviewer hopes that the authors will write this part more
Response: “So, the four extracts were tested as antiviral in vivo. Four extracts were administered orally to mice with different doses (3 times daily/7 days) following infection, 10 mg/kg is the most significant one [7].”
- Page 9, lines 309-310: “Baccharis dracunculifolia”
Write this in italics.
Response: “Baccharis dracunculifolia”
- Page 9, line 320: “coumaric acid”: Which coumaric acid, o-, p-, orp-?
Response: Kai et al. didn’t mention
- Page 10, Figure 4:“Flavinoids” should read “Flavonoids”.
Response: Adjusted
- Two structure formulas of “Flavonoids” and "Phenolic acids" are strange, because no phenolic hydroxyl group is written in them. Also, its more common to add a double bond to the side chain of the “Phenolic acids”.
Response: Adjusted
- Page 10, Table 2, title: “Propolis” should read “propolis”
(lowercase).
Response: Changed to “propolis”
- Page 10, Table 2, Item in the second column:“Compound name” should read “Compound names”.
Response: changed to “Compound names”
- Page 10, Table 2: The first four compounds (Melliferone, Moronic acid, Anwuweizonic acid, Betulonic Acid) are terpenes
Response: Adjusted
Concluded. 5. Propolis & COVID-19 (p.21-22)
- Page 22, lines 350-351: “rotavirus human rhinovirus so,” should read “rotavirus, human rhinovirus and so on,”
Response: Adjusted
- Page 22, line 353: “ --- different kinds DNA and RNA viruses ---” should read “ --- different kinds of DNA and RNA viruses ---”.
Response: "of" is added
- Page 22, lines 355-356: “ --- studied while chrysin and kaempferol, were ---” should read “ --- studied, while chrysin and kaempferol were ---”
Response: Adjusted
- Page 22, line 365: “75%, 63% and 44%” and “ -7.8 -7.8, and -7.6 kcal/mol”' Including other parts, unify to either "A, B and C" or "A, B, and C".
Response: Adjusted
- Page 22, line 371: “at -6.383, -6.097, -6.295 respectively [20].” should read “at -6.383, -6.097 and -6.295, respectively [20].”'
Response: Adjusted
- Page 22, line 387: “in the lungs” Put a period at the end of the sentence.
Response: Adjusted to “in the lungs.”
Immunomodulatory activity (page 23-)
- Page 23, lines 389-390: “Propolis has been mentioned as an immunomodulatory agent for centuries yet little was known about its action until the 1990s.” This should read “Alhough propolis has been mentioned as an immunomodulatory agent for centuries, little was known about its action until the 1990s.”'
Response: Adjusted
- Page 23, line 411: “the features characteristic of RA”“Features” and “characteristic” have the same meaning and they are
Response: “characteristic” is omitted
- Line 414: “propolis” should read “Brazilian propolis”
Response:" Brazilian" is added
- Page 23, line 415: “Th-17” should read “Th17”.
Response: Adjusted
- Page 23, line 430: “Toll-like receptors TLRs” should read “Toll-like
receptors (TLRs)”
Response: Adjusted
- Page 23, lines 430-441: “Toll-like receptors ------ and antimicrobial peptides [82]”. There is no description of propolis in this paragraph (12 lines).
Response: Thanks for the comment; in this section we have discussed the role of Toll-like receptors (TLRs) as one of the critical agent of the immune system for the recognition of microbes, the activation of the innate immune system and further initiation of the adaptive immune response
- Page 24, line 450: “Streptococcus mutans”Write this in italics.
Response: Adjusted
- Page 24, line 464: “administration” should read “Administration”.
Response: thanks for the comment but "administration" should be as it is, it comes after "Pontin et al."
- Page 24, line 466:” The authors” should read “Pontin et al.”
Response: "The authors" is replaced with "Pontin et al "
- Page 24, line 475: “ ---. As ---“ should read “ ---, because ---“
Response: "As" is replaced with "Because"
- Page 24, line 477: “(2017)” should read “[89]”
Response: Adjusted
-Page 24, line 481: “On the other hand NO levels ---” should read “On the other hand, NO levels ---”
Response: Adjusted
- Page 33, line 535, “7. Clinical applications of propolis as antiviral and immunomodulatory agent”: “agent” should read ”agents”.
Response: Adjusted
Table 3 (page 26, line 532~) The item name in the second column of Table 3 says “Type of extract/Major active constituents”, but “Major active constituents” are not written in Table3
Response: We have delayed "major active constituents" since some original papers lack this point
- Page 31, 33: Table 3:
Add an asterisk to the “NR” and “PBMNCs” in and below Table 3 (*1 and *2, respectively).
Response: Adjusted
- Page 34, lines 566-569 (Final paragraph of Chapter 7): Correct this partof English.
Response: We thank the referee for the positive comments. And we have addressed this issue as shown below;
“However, the propolis has plenty of biological and pharmacological properties, there is a lack of clinical reports on the effectiveness of propolis. Furthermore, some researches indicate that is the propolis unsafe due to it induces hypersensitivity, anaphylaxis, has adverse reactions such as allergic cheilitis, and oral ulceration (Table 4) [8].”
- Page 34, Table 4, title: “list of ---” should read “List of ---” 8: “ --- agent.” should read “ --- agents.”
Response: Adjusted
- Page 34, Table 4, The item name in the third column.:“no of participant” should read “Number of participants”.
Response: Adjusted
- Page 34, Table 4, below reference [122]: What is “ISRCTN17781274”? Doyou need this?
Response: “ISRCTN stood for 'International Standard Randomised Controlled Trial Number' ; ISRCTN17781274 is the reference of the stated clinical trial study.
In silico drug discovery
- Page 37, Table 5, title: “ --- for forty anti-viral compounds, darunavirand remdesivir ---”Actually, in Table 5, forty compounds, Darunavir and Favipiravir arelisted, but remdesivir is not. Which is correct, remdesvir orfavipiravir?
Response: The correct drug is favipiravir.
- Page 38, Table 5: “(E)-3-[4-hydroxy-“ should read “(E)-3-[4-Hydroxy-”.
Response: Adjusted
- Page 39, Table 5: “(E)-2,3-dihydro---” should read “(E)-2,3-Dihydro---”.
Response: Adjusted
- Page 44, Table 5: “2,2-dimethyl chromene-6---” should read “2,2-Dimethylchromene-6---”
Response: Adjusted.
- Page 45, Table 5: “m-coumaric acid” should read “m-Coumaric acid”.
Response: Adjusted
- Page 45, Table 5: “p-cumaric acid” should read “p-Coumaric acid”.
(misspelling: add “o”)
Response: Adjusted
- Page 46, Table 5: “Trans-cinnamic acid” should read “trans-Cinnamic
acid”.
Response: Adjusted
- Page 48 and 49, Figure 6: About color display:The two sets of color displays for the interactions listed below arealmost the same and cannot be distinguished.
- *Carbon Hydrogen Bond and Pi-Donor Hydrogen Bond
*Unfavorable Donor-Donor and Unfavorable Acceptor-Acceptor
*Alkyl and Pi-Alkyl
Response: The authors thank the reviewer for his comment. However, the color scale presented in Figure 6 is a standard scale defined by the Biovia software. It is also worth mentioning that the same color is used for the same class of interactions.
- Acknowledgments:
Line 710: “Authors are ---” should read “The authors are ---”
Response: "The" is added
Conflict of interest:
- Line 718: “Authors ---” should read “The authors ---”
Response: "The" is added

Reviewer 3 Report
The manuscript by Yosri et al describes the anti-viral and immunomodulatory properties of propolis, with an extensive description of chemical nature of compounds found in propolis of different origins, of studies about the anti-viral capacity of propolis and its components against different human viruses, including Covid-19 and a final in silico analysis of molecular docking of potential interaction of different propolis-bearing compounds against molecular targets of Sars-CoV2. The manuscript appears to be exhaustive and interesting, but major revisions are required. Apart from paragraphs 6 to 9 which are very well written, the initial paragraphs need extensive revision especially regarding English grammar and many mistyping errors. Many sentences have the wrong tense and some of them lack a verb. Other sentences have double subjects and should be divided in two separate phrases. Specific comments are detailed below.
Abstract
- Line 36. Please, correct as follows: “as a dietary supplementary and a folk remedy as well as it possesses”
- Line 42. Please, correct as follows: among which are terpenes…”
- Line 56. Eliminate “avoid”
Introduction
Here are some examples of the several mistakes in paragraphs from 1 to 5. Please, correct as follows and revise carefully the rest of manuscript.
- Line 79. resinous substance whose constituents…
- Line 80. plant materials…
- Line 82. in addition to keeping…
- Line 83. The word propolis in the Greek…
- Line 86. were recognized, since…
- Line 89. on the 17th (on must be lower case)
- Line 95. such as antiseptic, …
- Line 98. delete "wide of"
- Line 99. RNA viruses among which…
- Line 104. the main constituents of propolis are polyphenols
- Line 110. immunomodulatory drug against different viruses
- Line 114. This study sheds new light
- etc.
Other specific comments
- Line 134 and 173. Green, red and brown propolis should be defined in the Introduction or before citing them.
- Paragraph 3. This paragraph should start with a general description of categories/classes of compounds present in propolis with referring to Figure 1,2,3.
- Furthermore, Figure 1 should be subdivided according to flavonoid classes (e.g. flavonols, isoflavones)
- More precise citations of Figures 1,2,3 are needed. Otherwise, it is difficult to follow the description in paragraph 3.
- Pinocembrin and chrysin are not in Fig. 2 (Line 160).
- Drupanin, capillartemisin A, arthepillin and baccharin are not in Fig. 3 (Lines 165-166).
- Flavonoids, isoflavonoids (pratensein, violanone, formononetin, vestitone, and biochanin A) Figure?; (E)-3-[4-Hydroxy-3-(2-hydroxy-3-methylbut-3-en-1-yl)-5-(3-methylbut-2-en-1yl)phenyl] propenoic acid Figure?
- Title of Figure 3 is too generic... Figure 1 and 2 also include polyphenols
- Paragraph 4. It is very difficult to read and follow the description referring to Table 1. Organize Table 1 in the order descibed in the text. Also, define abbreviations used in the Table 1 and other tables.
- Lines 212-213. This sentence is unclear.
- Lines 216-217. This sentence lacks a verb
- Lines 218-221. This sentence lacks a verb and has two subjects
- Line 226. Viruses
- Lines 232-234. This sentence is unclear.
- Line 244. Incorrect verb
- Maximal suppuration… what does it mean?
- Lines 356-361. Divide in two sentences
Author Response
- The manuscript by Yosri et al describes the anti-viral and immunomodulatory properties of propolis, with an extensive description of chemical nature of compounds found in propolis of different origins, of studies about the anti-viral capacity of propolis and its components against different human viruses, including Covid-19 and a final in silico analysis of molecular docking of potential interaction of different propolis-bearing compounds against molecular targets of Sars-CoV2. The manuscript appears to be exhaustive and interesting, but major revisions are required. Apart from paragraphs 6 to 9 which are very well written, the initial paragraphs need extensive revision especially regarding English grammar and many mistyping errors. Many sentences have the wrong tense and some of them lack a verb. Other sentences have double subjects and should be divided in two separate phrases. Specific comments are detailed below.
Response: We would like to thank the reviewer for taking the time and effort necessary to provide such insightful comment; the manuscript has been carefully revised by native English speaker
Abstract
- Line 36. Please, correct as follows: “as a dietary supplementary and a folk remedy as well as it possesses”
Response: Adjusted
- Line 42. Please, correct as follows: among which are terpenes…”
Response: "them" is replaced with " which are"
- Line 56. Eliminate “avoid”
Response: Adjusted
Introduction
- Here are some examples of the several mistakes in paragraphs from 1 to 5. Please, correct as follows and revise carefully the rest of manuscript.
- Line 79. resinous substance whose constituents…
Response: "whose" is added
- Line 80. plant materials…
Response: Adjusted
- Line 82. in addition to keeping…
Response: Corrected
- Line 83. The word propolis in the Greek…
Response: "is" is replaced with "in"
- Line 86. were recognized, since…
Response: "where" is replaced with "since"
- Line 89. on the 17th (on must be lower case)
Response: Adjusted
- Line 95. such as antiseptic, …
Response: "such as" is added
- Line 98. delete "wide of"
Response: Adjusted
- Line 99. RNA viruses among which…
Response: "them" is replaced with "which"
- Line 104. the main constituents of propolis are polyphenols
Response: Corrected
- Line 110. immunomodulatory drug against different viruses
Response: "lead" is omitted
- Line 114. This study sheds new light
Response: shines is replaced with "sheds"
Other specific comments
- Line 134 and 173. Green, red and brown propolis should be defined in the Introduction or before citing them.
Response: Adjusted
“Brazilian Green propolis has also an anti-inflammatory, anti-bacterial, and anti-ulcer properties in traditional medicine [34]..”
“Saito et al. evaluated conventional (Soxhlet) extraction with ethanol, methanol, water, and hexane as solvents and supercritical CO2 extraction for 2 different types of (green and red) propolis from Brazil, and found that methanolic extraction was the best method in terms of yield for both propolis, while fractionation of ethanolic extract of the red propolis with supercritical CO2 produced fractionated extracts with much higher flavonoid contents than those found in the original ethanolic extract [39].
- Paragraph 3. This paragraph should start with a general description of categories/classes of compounds present in propolis with referring to Figure 1,2,3.
Response: "Propolis is very rich in polyphenolic compounds mainly flavonoids and cinnamic acids (Figures 1-3)" is added
- Furthermore, Figure 1 should be subdivided according to flavonoid classes (e.g. flavonols, isoflavones)
Response: Figure 1 is subdivided to: Flavanones, Flavones, Isoflavones, Isoflavanones, Chalcone, Flavanol, Isoflavan
- More precise citations of Figures 1,2,3 are needed. Otherwise, it is difficult to follow the description in paragraph 3.
Response: Adjusted
- Pinocembrin and chrysin are not in Fig. 2 (Line 160).
Response: "Fig. 2" is adjusted to "Figures 1-2"
- Drupanin, capillartemisin A, arthepillin and baccharin are not in Fig. 3 (Lines 165-166).
Response: "Figure 3" is adjusted to "Figure 2"
- Flavonoids, isoflavonoids (pratensein, violanone, formononetin, vestitone, and biochanin A) Figure?; (E)-3-[4-Hydroxy-3-(2-hydroxy-3-methylbut-3-en-1-yl)-5-(3-methylbut-2-en-1yl)phenyl] propenoic acid Figure?
Response: “flavonoids, isoflavonoids (pratensein, violanone, formononetin, vestitone, and biochanin A) (Figure 1); di- and triterpenoids, as well new compound; (E)-3-[4-hydroxy-3-(2-hydroxy-3-methylbut-3-en-1-yl)-5-(3-methylbut-2-en-1-yl)phenyl] propenoic acid among other constituents (Figure 3) [2]”
- Title of Figure 3 is too generic... Figure 1 and 2 also include polyphenols
Response: The title of figure 3 is changed to " Figure 3. Common ester and acid compounds are isolated and identified from propolis of Apis mellifera L.
- Paragraph 4. It is very difficult to read and follow the description referring to Table 1. Organize Table 1 in the order descibed in the text. Also, define abbreviations used in the Table 1 and other tables.
Response: Adjusted
- Lines 212-213. This sentence is unclear.
Response: the sentence is adjusted to " Propolis has long been used to treat various diseases in particular viral infections and more recently COVID-19 [43]. ".
- Lines 216-217. This sentence lacks a verb
Response: The sentence is adjusted to " the exploring of novel candidates or compounds with different anti-viral mechanisms from safe and effective naturally sources are urgently needed.".
- Lines 218-221. This sentence lacks a verb and has two subjects
Response: the sentences is adjusted to " Herpes simplex viruses (HSV) types 1 and 2 are considered the most prevalent human pathogens that cause watery blisters on the skin or mucosae. "
- Line 226. Viruses
Response: Adjusted
- Lines 232-234. This sentence is unclear.
Response: “In addition, Brazilian hydroalcoholic brown propolis extract (HPE) was reported for the first time for its protective activity against vaginal lesions induced by HSV-2 in a female BALB/c mouse model.”
- Line 244. Incorrect verb
Response: “added”
- Maximal suppuration… what does it mean?
Response: Modified to “The maximum dose of propolis at 66.6 µg/mL exhibited anti-HIV-1 viral activity with 85 and 98% for infect cells CD4+ and microglial cell cultures, respectively through inhibition of viral entry and reverse transcriptase [51]. “
- Lines 356-361. Divide in two sentences
Response: Adjusted

Round 2
Reviewer 1 Report
In my opinion the manuscript may be published in Foods
Author Response
Comments and Suggestions for Authors
In my opinion the manuscript may be published in Foods
Response: We would like to thank the reviewer for the nice words.
Reviewer 2 Report
Reviewer's comments on the corrected manuscript
Most of the points in this corrected manuscript have been corrected. However, some points have not been corrected. Also, new mistakes can be seen. This reviewer points out them below.
About chapter numbers:
There are new mistakes in the corrected manuscript regarding the chapter numbers in the main text. The chapter numbers in the original manuscript were correct. That is, there are two chapters. "5. Propolis & COVID-19" and "5. Immunomodulatory activity". Please correct the second chapter "5. Immunomodulatory activity" to "6. Immunomodulatory activity". Then, move chapters 6, 7, and 8 to chapters 7, 8, and 9, respectively.
Page 4, line 162-164: “Furthermore, 2 flavonoids (pinocembrin and chrysin) (Figure 1), trans-cinnamic acid, and 4 phenolic cinnamic acid (caffeic acid, p-coumaric acid, ferulic acid, and m-coumaric acid) (Figure 2) besides many of volatile compounds had been identified in 2 propolis samples and are different in apiary geographical locations in Italy via HPLC-UV, NMR, and GC–MS, respectively [1].”
According to the reference [1], the content of 2 flavonoids (pinocembrin and chrysin) and caffeic acid in propolis samples obtained from two locations, highlands and plains, was not different between the two locations. It was reported that their statistic evaluation (p-value) was not significant.
Therefore, the reviewer proposes to modify the text as follows:
“Furthermore, 2 flavonoids (pinocembrin and chrysin) (Figure 1), trans-cinnamic acid, and 4 phenolic cinnamic acid (caffeic acid, p-coumaric acid, ferulic acid, and m-coumaric acid) (Figure 2) besides many of volatile compounds had been identified, by HPLC-UV, NMR, and GC–MS, from 2 propolis samples from two apiaries with different geographical locations in Italy [1].”
Page 4, lines 166-168: “--- including prenylated phenylpropanoids (drupanin, capillartemisin A, (E)-2,3-dihydroconiferyl p-coumarate, 2,2-dimethylchromene-6-propenoic acid, artepillin C and baccharin) (Figure 2); ”
(E)-2,3-dihydroconiferyl p-coumarate is not a prenylated phenylpropanoid. Therefore, that part should read “--- including prenylated phenylpropanoids (drupanin, capillartemisin A, 2,2-dimethylchromene-6-propenoic acid, artepillin C and baccharin) and (E)-2,3-dihydroconiferyl p-coumarate, (Figure 2);”.
Pages 6 and 7, Figures 2 and 3: Structural formulas of methyl groups in some compounds.
The writing of prenyl groups has been improved. However, in the case of writing a methyl group, it is inconsistent whether it is CH3 or it is an abbreviated bar. It is still inconsistent in the Figure 2. Moreover, it is inconsistent between Figures 2 and 3.
Page 7, Figure 3; Page 40, Table 5, No 4; (in three places):
“-2-en-1yl)phenyl]” should read “-2-en-1-yl)phenyl]” [Add a hyphen between 1 and yl.]
Page 8, line 248: “Another study conducted on propolis originating from Canada.” should read “Another study was conducted on propolis originating from Canada.”
Page 8, line 252: “the France propolis” should read “the French propolis”.
Page 9, line 297-298: “--- HRV-4, respectively, while, p-coumaric acid shows ---” should read “--- HRV-4, respectively, while p-coumaric acid shows ---”. (Delete the comma after “while”)
Page 10, Figure 4: Actually the reviewer’s comment was not adjusted.
Two structure formulas of “Flavonoids” and "Phenolic acids" are strange, because no phenolic hydroxyl group is written in them. Also, it is more common to add a double bond to the side chain of the “Phenolic acids”.
Page 10, Table 2: The first four compounds (Melliferone, Moronic acid, Anwuweizonic acid, Betulonic Acid) are terpenes.
The author’s answer is adjusted. But, how to adjust in the text or Figure/Table?
Page 22, line 379: “-and 6.295,” should read “and -6.295,”
Page 24, line 472: “--- by Pontin et al. administration ---”:
How about such a correction: “--- by Pontin et al., that is, administration ---”.
Page 24, line 489: “On the other hand NO (nitric oxide) levels ---” should read “On the other hand, NO (nitric oxide) levels ---”. Insert a comma between “hand” and “NO”.
Page 34, lines 601-606 (Final paragraph of Chapter 7):
The authors corrected this paragraph as follows:
“However, the propolis has plenty of biological and pharmacological properties, there is a lack of clinical reports on the effectiveness of propolis. Furthermore, some researches indicate that is the propolis unsafe due to it induces hypersensitivity, anaphylaxis, has adverse reactions such as allergic cheilitis, and oral ulceration (Table 4) [126]”.
The reviewer proposes the following revisions.
“Although the propolis has plenty of biological and pharmacological properties, there is a lack of clinical reports on the effectiveness of propolis. Furthermore, some researches indicated that the propolis is unsafe because it induces hypersensitivity such as anaphylaxis, and had adverse reactions such as allergic cheilitis and oral ulceration (Table 4) [126]”.
Page 34, Table 4, title (line 607): “--- agent.” should read “--- agents.”
Page 37, Table 5, title: “--- for forty anti-viral compounds, darunavir and remdesivir ---”
Actually, in Table 5, forty compounds, Darunavir and Favipiravir are listed, but remdesivir is not. Which is correct, remdesvir or favipiravir?
The authors response was “The correct drug is favipiravir.” So, in the title “remdesivir” should read “favipiravir”.
Pages 40 and 42, Table 5, Compound names, No. 4, 6, 11, 12, and 32:
The line break position in the middle of a word is not appropriate. Or add a hyphen at the line break position.
Page 50, Author Contributions: “writing—“ should read “”Writing—
Concluded.
Author Response
- Most of the points in this corrected manuscript have been corrected. However, some points have not been corrected. Also, new mistakes can be seen. This reviewer points out them below.
Response: We would like to thank the reviewer for taking the time and effort necessary to provide such insightful comment; the manuscript has been carefully revised
About chapter numbers:
- There are new mistakes in the corrected manuscript regarding the chapter numbers in the main text. The chapter numbers in the original manuscript were correct. That is, there are two chapters. "5. Propolis & COVID-19" and "5. Immunomodulatory activity". Please correct the second chapter "5. Immunomodulatory activity" to "6. Immunomodulatory activity". Then, move chapters 6, 7, and 8 to chapters 7, 8, and 9, respectively.
Response: Adjusted
“5. Propolis as a treatment for COVID-19”
“6. Immunomodulatory activity”
“7. Clinical applications of propolis as an antiviral and immunomodulatory agents “
“8. In silico drug discovery “
“9. Conclusion”
- Page 4, line 162-164: “Furthermore, 2 flavonoids (pinocembrin and chrysin) (Figure 1), trans-cinnamic acid, and 4 phenolic cinnamic acid (caffeic acid, p-coumaric acid, ferulic acid, and m-coumaric acid) (Figure 2) besides many of volatile compounds had been identified in 2 propolis samples and are different in apiary geographical locations in Italy via HPLC-UV, NMR, and GC–MS, respectively [1].” According to the reference [1], the content of 2 flavonoids (pinocembrin and chrysin) and caffeic acid in propolis samples obtained from two locations, highlands and plains, was not different between the two locations. It was reported that their statistic evaluation (p-value) was not significant.
Therefore, the reviewer proposes to modify the text as follows:
“Furthermore, 2 flavonoids (pinocembrin and chrysin) (Figure 1), trans-cinnamic acid, and 4 phenolic cinnamic acid (caffeic acid, p-coumaric acid, ferulic acid, and m-coumaric acid) (Figure 2) besides many of volatile compounds had been identified, by HPLC-UV, NMR, and GC–MS, from 2 propolis samples from two apiaries with different geographical locations in Italy [1].”
Response: Adjusted to “Furthermore, 2 flavonoids (pinocembrin and chrysin) (Figure 1), trans-cinnamic acid, and 4 phenolic cinnamic acid (caffeic acid, p-coumaric acid, ferulic acid, and m-coumaric acid) (Figure 2) besides many of volatile compounds had been identified, by high pressure liquid chromatography with UV detector (HPLC-UV), nuclear magnetic resonance (NMR), and gas chromatography – mass spectrometry (GC–MS), from 2 propolis samples from two apiaries with different geographical locations in Italy [1].”
- Page 4, lines 166-168: “--- including prenylated phenylpropanoids (drupanin, capillartemisin A, (E)-2,3-dihydroconiferyl p-coumarate, 2,2-dimethylchromene-6-propenoic acid, artepillin C and baccharin) (Figure 2); ”
(E)-2,3-dihydroconiferyl p-coumarate is not a prenylated phenylpropanoid. Therefore, that part should read “--- including prenylated phenylpropanoids (drupanin, capillartemisin A, 2,2-dimethylchromene-6-propenoic acid, artepillin C and baccharin) and (E)-2,3-dihydroconiferyl p-coumarate, (Figure 2);”.
Response: Done
“--- including prenylated phenylpropanoids (drupanin, capillartemisin A, 2,2-dimethylchromene-6-propenoic acid, artepillin C and baccharin) and (E)-2,3-dihydroconiferyl p-coumarate, (Figure 2);”
- Pages 6 and 7, Figures 2 and 3: Structural formulas of methyl groups in some compounds.
The writing of prenyl groups has been improved. However, in the case of writing a methyl group, it is inconsistent whether it is CH3 or it is an abbreviated bar. It is still inconsistent in the Figure 2. Moreover, it is inconsistent between Figures 2 and 3.
Response: Adjusted.
- Page 7, Figure 3; Page 40, Table 5, No 4; (in three places):
“-2-en-1yl)phenyl]” should read “-2-en-1-yl)phenyl]” [Add a hyphen between 1 and yl.]
Response: Changed to “(E)-3-[4−Hydroxy-3-(2-hydroxy-3- methylbut-3-en-l-yl)-5-(3-methylbut-2-en-l-yl)phenyl] propenoic acid”
- Page 8, line 248: “Another study conducted on propolis originating from Canada.” should read “Another study was conducted on propolis originating from Canada.”
Response: Changed to “Another study was conducted on propolis originating from Canada”
- Page 8, line 252: “the France propolis” should read “the French propolis”.
Response: Adjusted to “the French propolis”
- Page 9, line 297-298: “--- HRV-4, respectively, while, p-coumaric acid shows ---” should read “--- HRV-4, respectively, while p-coumaric acid shows ---”. (Delete the comma after “while”)
Response: Done
“HRV-4, respectively, while p-coumaric acid shows”
- Page 10, Figure 4: Actually the reviewer’s comment was not adjusted.
Two structure formulas of “Flavonoids” and "Phenolic acids" are strange, because no phenolic hydroxyl group is written in them. Also, it is more common to add a double bond to the side chain of the “Phenolic acids”.
Response: Adjusted
- Page 10, Table 2: The first four compounds (Melliferone, Moronic acid, Anwuweizonic acid, Betulonic Acid) are terpenes.
The author’s answer is adjusted. But, how to adjust in the text or Figure/Table?
Response: “Additionally, eight compounds isolated from Brazilian propolis, including four triterpenoids (melliferone, moronic acid, anwuweizonic acid, and betulonic acid) and….”
- Page 22, line 379: “-and 6.295,” should read “and -6.295,”
Response: Adjusted
- Page 24, line 472: “--- by Pontin et al. administration ---”:
How about such a correction: “--- by Pontin et al., that is, administration ---”.
Response: Modified
“by Pontin et al. that is, administration”
- Page 24, line 489: “On the other hand NO (nitric oxide) levels ---” should read “On the other hand, NO (nitric oxide) levels ---”. Insert a comma between “hand” and “NO”.
Response: Done
“On the other hand, nitric oxide (NO) levels”
- Page 34, lines 601-606 (Final paragraph of Chapter 7):
The authors corrected this paragraph as follows:
“However, the propolis has plenty of biological and pharmacological properties, there is a lack of clinical reports on the effectiveness of propolis. Furthermore, some researches indicate that is the propolis unsafe due to it induces hypersensitivity, anaphylaxis, has adverse reactions such as allergic cheilitis, and oral ulceration (Table 4) [126]”.
The reviewer proposes the following revisions.
“Although the propolis has plenty of biological and pharmacological properties, there is a lack of clinical reports on the effectiveness of propolis. Furthermore, some researches indicated that the propolis is unsafe because it induces hypersensitivity such as anaphylaxis, and had adverse reactions such as allergic cheilitis and oral ulceration (Table 4) [126]”.
Response: “Although the propolis has plenty of biological and pharmacological properties, there is a lack of clinical reports on the effectiveness of propolis. Furthermore, some researches indicated that the propolis is unsafe because it induces hypersensitivity such as anaphylaxis, and had adverse reactions such as allergic cheilitis and oral ulceration (Table 4)”
- Page 34, Table 4, title (line 607): “--- agent.” should read “--- agents.”
Response: “Table 4: List of clinical application of propolis as anti-viral and immunomodulatory agents.”
- Page 37, Table 5, title: “--- for forty anti-viral compounds, darunavir and remdesivir ---”
Actually, in Table 5, forty compounds, Darunavir and Favipiravir are listed, but remdesivir is not. Which is correct, remdesvir or favipiravir?
The authors response was “The correct drug is favipiravir.” So, in the title “remdesivir” should read “favipiravir”.
Response: Changed to “favipiravir”
- Pages 40 and 42, Table 5, Compound names, No. 4, 6, 11, 12, and 32:
The line break position in the middle of a word is not appropriate. Or add a hyphen at the line break position.
Response: Adjusted
- Page 50, Author Contributions: “writing—“ should read “”Writing—
Response: Done
- Conclude
Response: this section rewritten
“Propolis has been used for centuries as a natural remedy, and modern laboratory investigations have confirmed the effectiveness of propolis extracts and derivatives against multiple disease models, including viral infections. These therapeutic effects are attributable to high concentrations of bioactive flavonoids, phenolic acids, and esters that target a myriad of pathological and reparative processes, including immune signaling pathways. Demonstrated efficacy against a wide range of human viruses has provided a rationale for studies on the efficacy of propolis against SARS-COV-2. Further, 40 propolis derivatives have shown high affinity binding in silico SARS-CoV-2 proteins and the human target ACE2, with one, retusapurpurin A, demonstrating particularly potent binding to SARS-CoV-2 3CLpro, RdRp, ACE2, RBD, and NSP13 inhibitor. These results suggest that retusapurpurin A and other components such as baccharin are promising therapeutic candidates for COVID-19 treatment.
In addition to viral infection, propolis has anti-inflammatory and immunomodulatory activities that have proven effective in models of cancer and immune-related diseases such as celiac disease, uveoretinitis, allergic asthma, and rheumatoid arthritis. While there have been few studies on the clinical efficacy of propolis preparations against human diseases, clinical trials are feasible due to the generally good safety profiles of propolis derivatives.”

Reviewer 3 Report
The manuscript by Yosri et al was significantly improved and all revisions requested were addressed.
Author Response
The manuscript by Yosri et al was significantly improved and all revisions requested were addressed.
Response: We would like to thank the reviewer for taking the time and effort.